# A two-level staging system for the embryonic morphogenesis of the Mediterranean fruit fly (medfly) *Ceratitis capitata*

**Frederic Strobl**[1]*, **Alexander Schmitz**[1], **Marc F. Schetelig**[2], **Ernst H. K. Stelzer**[1]

**1** Physical Biology / Physikalische Biologie (IZN, FB 15), Buchmann Institute for Molecular Life Sciences (BMLS), Cluster of Excellence Frankfurt–Macromolecular Complexes (CEF–MC), Goethe-Universität–Frankfurt am Main (Campus Riedberg), Frankfurt am Main, Germany, **2** Department of Insect Biotechnology in Plant Protection, Liebig Centre for Agroecology and Climate Impact Research, Justus-Liebig-Universität Gießen, Gießen, Germany

* frederic.strobl@physikalischebiologie.de

**Data Availability Statement:** All dataset files are available from the Zenodo database (accession number(s) https://doi.org/10.5281/zenodo.

## Abstract

Comparative studies across multiple species provide valuable insights into the evolutionary diversification of developmental strategies. While the fruit fly *Drosophila melanogaster* has long been the primary insect model organism for understanding molecular genetics and embryonic development, the Mediterranean fruit fly *Ceratitis capitata*, also known as medfly, presents a promising complementary model for studying developmental biology. With its sequenced genome and a diverse array of molecular techniques, the medfly is well-equipped for study. However, an integrative framework for studying its embryogenesis is currently lacking. In this study, we present a two-level staging system for the medfly based on nine datasets recorded using light sheet fluorescence microscopy. The upper level features of six consecutive embryogenetic events, facilitating comparisons between insect orders, while the lower level consists of seventeen stages, adapted from the fruit fly, allowing for comparisons within the Diptera. We provide detailed descriptions of all identifiable characteristics in multiple formats, including a detailed timetable, comprehensively illustrated figures for all embryogenetic events, glossary-like tables for selected structures and processes, as well as a stage-based quick lookup chart. One remarkable difference between the fruit fly and the medfly is that in the latter, the amnioserosa differentiates and unfolds already during gastrulation. Our staging system, which is based on systematically acquired fluorescence live imaging data, provides standard deviations for each developmental time point and serves as a template for future studies seeking to establish an integrative morphogenic framework for other emerging model insect species.

## Introduction

For over a century, the fruit fly *Drosophila melanogaster* has been the primary insect model organism for developmental biology and genetic studies [1, 2]. However, a comparative approach, *i.e.*, identification and classification of similarities and differences across species, is

6448019 (DS0001), https://doi.org/10.5281/zenodo.6448433 (DS0002), https://doi.org/10.5281/zenodo.6450798 (DS0003), https://doi.org/10.5281/zenodo.6451099 (DS0004), https://doi.org/10.5281/zenodo.6453980 (DS0005), https://doi.org/10.5281/zenodo.6455038 (DS0006), https://doi.org/10.5281/zenodo.6456820 (DS0007), https://doi.org/10.5281/zenodo.6457004 (DS00008), https://doi.org/10.5281/zenodo.6457894 (DS0009), https://doi.org/10.5281/zenodo.13842879 (Mathematica workbooks, intermediate data, visualizations) or via the associated data publication (accession number https://doi.org/10.1038/s41597-022-01443-x).

**Funding:** The author(s) received no specific funding for this work.

**Competing interests:** The authors have declared that no competing interests exist.

essential for gaining insights into the evolutionary diversification of developmental strategies and the mechanisms that drive these processes [3]. Over the past few decades, there has been an increase in the number of insect species used as model organisms for evolutionary developmental biology, *e.g.*, the squinting bush brown *Bicyclus anynana* [4] among the Lepidoptera, the red flour beetle *Tribolium castaneum* among the Coleoptera [5], the honeybee *Apis mellifera* [6] as well as the jewel wasp *Nasonia vitripennis* [7] among the Hymenoptera, and the two-spotted cricket *Gryllus bimaculatus* among the Orthoptera [8]. Within the Diptera, the scuttle fly *Megaselia abdita* [9] and the moth midge *Clogmia albipunctata* [10] are used as 'phylogenetic links' between the fruit fly and members from other insect orders. However, the fruit fly differs significantly from the scuttle fly and the moth midge in many developmental features due to the relatively large phylogenetic distance of approximately 150 and 250 million years [11, 12], respectively. One remarkable morphogenic difference lies within the formation of extra-embryonic membranes, with fruit fly embryos developing only one dorsally located membrane, the amnioserosa [13]. In contrast, embryos of the scuttle fly and moth midge are covered by the amnion on their dorsal respectively ventral side and completely engulfed by the serosa [11, 12, 14–17].

The Mediterranean fruit fly *Ceratitis capitata*, commonly known as the medfly, is a promising candidate for comparison with the fruit fly, as the two species descended from a common ancestor around 80 million years ago [18–20], creating a much shorter phylogenetic distance than that between the fruit fly and either the scuttle fly or the moth midge. In addition, as a well-established model organism in agricultural and pest-associated research [21], the medfly has a fully sequenced genome [22] and a large arsenal of molecular techniques is available for developmental studies. These features render it a strong choice for investigating morphogenic processes and gaining insights into the evolution of developmental processes. Juxtaposing species such as the fruit fly and the medfly provides researchers with a clearer understanding of both shared and unique developmental processes within the Diptera, and specifically within the Schizophora.

Certain aspects of medfly development have already been investigated. For example, specific morphologic structures of the adult medfly, such as the head and abdomen [23], copulation site [24], reproductive accessory glands [25], and antennal lobes [26], have been studied. In terms of embryonic development, the spatiotemporal expression patterns of genes such as *otd* [27], *nos* [28], *slbo*, *grk*, *dpp*, *tkv*, *mirr*, *pip*, *br* [29] or *twit* and other factors of the Ly6 gene family [30] have been documented. In addition, multiple transcriptome profiles have been analyzed [31–34]. However, besides the early synchronous division waves [35], cellularization [36] and germ cell formation [37], only limited information regarding the embryonic morphogenesis of medfly is available. In consequence, a comprehensive staging system for medfly embryonic development is lacking, which render it difficult to integrate the diverse data in a spatiotemporally organized context. This contrasts not only with the fruit fly as the best-established insect model organism [38], but also with the aforementioned scuttle fly [12] and moth midge [11] as well as other Nematoceran species such as the predatory gall midge *Aphidoletes aphidimyza* [39], and the yellow fever mosquito *Aedes aegypti* [40, 41].

Our study establishes such an integrative framework for the medfly based on long-term fluorescence live imaging data [42] from transgenic embryos that ubiquitously and constitutively express nuclear-localized EGFP [43]. Of the nine available datasets, which were recorded using light sheet fluorescence microscopy (LSFM) [44–46], six show the embryo *in toto* along four orientations for approximately 97% of the entire embryonic development. These datasets enable us to establish a two-level staging system featuring a detailed, comprehensive, and well-illustrated description of (i) six general embryogenetic events and (ii) seventeen stages adapted from the fruit fly framework [38]. The remaining three datasets cover shorter periods at a

higher level of detail and were used to illustrate specific aspects of development such as pole cell formation and the morphogenic movements during head involution. Since the long-term live imaging datasets provide a temporal resolution representing less than one percent of the total embryonic development, robust standard deviations are also provided. Additionally, we provide a list of structures and processes known in related species for which we found no evidence in the medfly, either due to limitations in the data, or since they are nonexistent in the medfly. Taken together, the work is a valuable resource for future studies on medfly embryogenesis and significantly contributes to the morphogenic aspect of comparative insect developmental biology.

## Materials and methods

### Datasets background and quality control

The fluorescence microscopy data presented in this study derive from the homozygous *Ceratitis capitata* (Wiedemann) *TREhs43-hid*$^{Ala5}$_F1m2 transgenic line [43] that expresses nuclear localized EGFP under control of the fruit fly *polyubiquitin* promoter. Data acquisition using LSFM, including microscope setup, embryo preparation, live imaging, larva retrieval, preliminary image processing, and temporal alignment of the nine available long-term live imaging datasets is explained in the respective peer-reviewed data publication [42]. Conventionally, we define three microscopy axes as follows: the illumination axis as $x$, the rotation axis as $y$, and the detection axis as $z$ [47]. In six datasets (DS0001–DS0006), the embryo was acquired *in toto* along the four orientations 0˚, 90˚, 180˚ and 270˚ under similar conditions for up to 65 hours, covering approximately 97% of the entire embryogenesis period, essentially missing only very early stages. The embryo of dataset DS0006 did not develop into a functional adult and was excluded from further analysis. The embryos of datasets DS0001 and DS0003 were imaged along the ventrolateral-dorsolateral axes, whereas the embryos of datasets DS0002, DS0004, and DS0005 were imaged along the ventral-dorsal and lateral axes. The remaining three datasets (DS0007–DS0009) cover specific aspects of development, *i.e.*, pole cell formation (DS0007) and head involution (DS0008 and DS0009).

### Fluorescence microscopy data processing and analysis

Microscopy data were processed (*e.g.*, rotation, cropping) using Fiji [48]. Analysis was performed using Mathematica (Version 13.3.0.0, Wolfram Research, Champaign, Illinois, USA). The Mathematica-based data processing and analysis code (in the form of Mathematica notebooks) as well as partially and or fully processed image data are available for download at Zenodo https://www.doi.org/10.5281/zenodo.13842879. Please note that, for privacy reasons, file paths referencing internal data storage servers have been removed from the notebooks.

### Length and speed measurements

Embryo length as a function of time over the complete embryonic development period was calculated only from datasets DS0001–DS0003 since in DS0004 and DS0005, the embryos protruded from the volume of view along $y$. The $z$ maximum projections for datasets DS0001–DS0003 were slightly rotated to align the anterior-posterior axis of the embryo with $y$. At each time point, the following algorithm was applied to compute the embryo length: Firstly, the image was cropped to the region of interest containing the embryo. Secondly, mean filtering with a 3×3 pixel range was applied to smooth the image. Thirdly, a local thresholding algorithm is used to obtain a binary image. Pixel values above the mean intensity computed in a range of 150×150 pixels were set to 1 and others with 0. Fourthly, the largest object in the

obtained binary image was retrieved, and the convex hull of this region was computed. Fifthly, the largest extension of the convex hull along *y* was computed as the length of the embryo. The distance and movement speed of the germband tip along the dorsal side towards the anterior pole during stages 8, 9, 10, and 11 were determined by manually locating the posterior tip of the germband in *z* maximum projections along the dorsolateral or dorsal orientation in datasets DS0001–DS0003.

### Nuclei area and density measurements

The average cell nuclei area and density at three time points during blastoderm formation were calculated in the *z* maximum projections: Firstly, the embryo was segmented with Otsu's algorithm [49] and the position of the posterior end of the embryo is determined. Secondly, at a distance of 150 pixels from the posterior end of the embryo, a 150×150 pixel area was extracted from the image. Gaussian filtering with a 2×2 pixel range was applied followed by local thresholding to obtain a rough segmentation of cell nuclei. The local threshold range was set to 15 for the first two time points and to 10 for the latter time point. Thirdly, pixel values above the determined local mean intensity were set to 1 and others with 0, and morphological filling was used to remove holes in the segmented area. Fourthly, to separate clustered cell nuclei, a seeded immersion watershed algorithm [50, 51] was applied to obtain individually labeled cell nuclei. Seed points were identified using a multi-scale Laplacian of Gaussian filtering. The minimum and maximum range of the Laplacian of Gaussian filter were set to 5, 15 for the former two time points and 1, 10 for the latter time point. Fifthly, the immersion watershed algorithm was initialized at the identified seed locations and applied to the inverted Gaussian filtered image. The final segmentation of cell nuclei was obtained by multiplying the local thresholding result with the immersion watershed output. Objects that were connected to the image border were discarded, and the number of cell nuclei and the area of each segmented cell nucleus was computed.

### Pole cell nuclei counting

The pole cells were counted in the *z* stacks in DS0001 and DS0002 based on TP0010, in DS0003 and DS0004 based on TP0009 and in DS0007 based on TP0007 and TP0009: Firstly, the *z* stacks were cropped to the region of interest and rotated around *y* by -40° for DS0001, -7° for DS0002, +41° for DS0003 and +13° for DS0004 to align the embryonal axes. Secondly, Laplacian of Gaussian filtering with a with a 3×3 pixel range was applied followed by local thresholding to obtain a rough segmentation of cell nuclei. Thirdly, pixel values above the determined local mean intensity were set to 1 and others with 0. Holes in the obtained segmentation were removed using morphological filling. Fourthly, to distinguish clustered cell nuclei, a seeded immersion watershed algorithm [50, 51] was applied to obtain individually labeled cell nuclei. Seed points were identified using a multi-scale Laplacian of Gaussian filtering. Fifthly, the immersion watershed algorithm was initialized at the determined seed locations and applied to the inverted Gaussian filtered image. The final segmentation of cell nuclei was obtained by multiplying the local thresholding result with the immersion watershed.

## Results

### A two-level staging system for the embryonic morphogenesis of the medfly

A well-defined staging system is fundamental for a comprehensive description of embryonic morphogenesis. The nine datasets were already temporally aligned in the accompanying data publication [42], this preliminary concept is expanded upon in this study. The rationale for a

two-level staging is as follows: The comparative approach, *i.e.*, juxtaposing and contrasting the embryonic morphogenesis of multiple species, embeds developmental biology into an evolutionary background and provides insights that are not obtainable when species are analyzed isolated from their phylogenetic lineage. The comparative approach is supported by an increase in the number of insect model organisms over the last decade. For distantly related species, only general equivalents can be identified, but closely related species typically exhibit a high degree of similarity to the embryonic morphogenesis of the fruit fly even with regard to highly specific processes. Hence, a two-level staging system that incorporates general embryogenetic events on the upper level and species-specific stages on the lower level is proposed.

The upper level serves primarily to facilitate morphogenic comparison of the medfly with species belonging to other insect orders, whereas the lower level allows the comparison within the Diptera. Extensive literature research on morphologic and morphogenic insect embryogenesis studies published during the last 40 years revealed six prominent consecutive embryogenetic events (S1 Table), which we define as the upper level and refer to with Roman numerals: (I) blastoderm formation, (II) early gastrulation, (III) germband elongation, (IV) germband retraction, (V) dorsal closure and (VI) muscular movement. Since there is no conventional designation for two of these events, we defined the period of gastrulation that lasts until the onset of germband elongation as 'early gastrulation' and the period spanning from the end of dorsal closure to the time point when the embryo hatches as 'muscular movement'. Although these events are not evident in all listed species, they recur at high frequency.

The lower level is defined by highly characteristic morphological states or morphogenic processes. During the evaluation of the image data, we found numerous similarities to the fruit fly (*e.g.*, cephalic furrow formation during early gastrulation or head involution during dorsal closure) and therefore decided to adopt the seventeen stages.

## Quantification of certain aspects of medfly embryogenesis

Due to the intrinsic properties of light sheet fluorescence microscopy, the respective data are well-suited for general and specific quantitative analyses. The temporal alignment of DS0001–DS0005 allows comparison in terms of absolute embryonic development time (Fig 1A) and relative completion of embryonic development (Fig 1B), as well as the calculation of the respective deviations. Deviational peaks occur at the transition from stage 10 to stage 11 (*i.e.*, at the end of germband elongation), during stage 13 (*i.e.*, at the beginning of dorsal closure), and around stages 16 and 17 (*i.e.*, during muscular movement).

Furthermore, temporal alignment in conjunction with image segmentation allowed us to measure the extent of the embryo along the anterior-posterior axis as a function of time (Fig 1C). The change in length varies substantially over the course of embryogenesis, resulting in multiple relative maxima and minima. Whenever appropriate, changes and peaks are discussed in the descriptive parts of the result and discussion sections below.

## Description of medfly embryogenesis

Description of medfly embryogenesis spans from the first withdrawal of the yolk during blastoderm formation to the moment when the larva hatches from the egg and is structured into several complementary elements:

- The running text is divided into six sections that relate to the six embryogenetic events. Each section is accompanied by a thoroughly illustrated figure. Complex processes involving multiple structures are described over longer periods. Only medfly-related studies are referenced within the running text. Circled numbers refer to arrows in the respective figures.

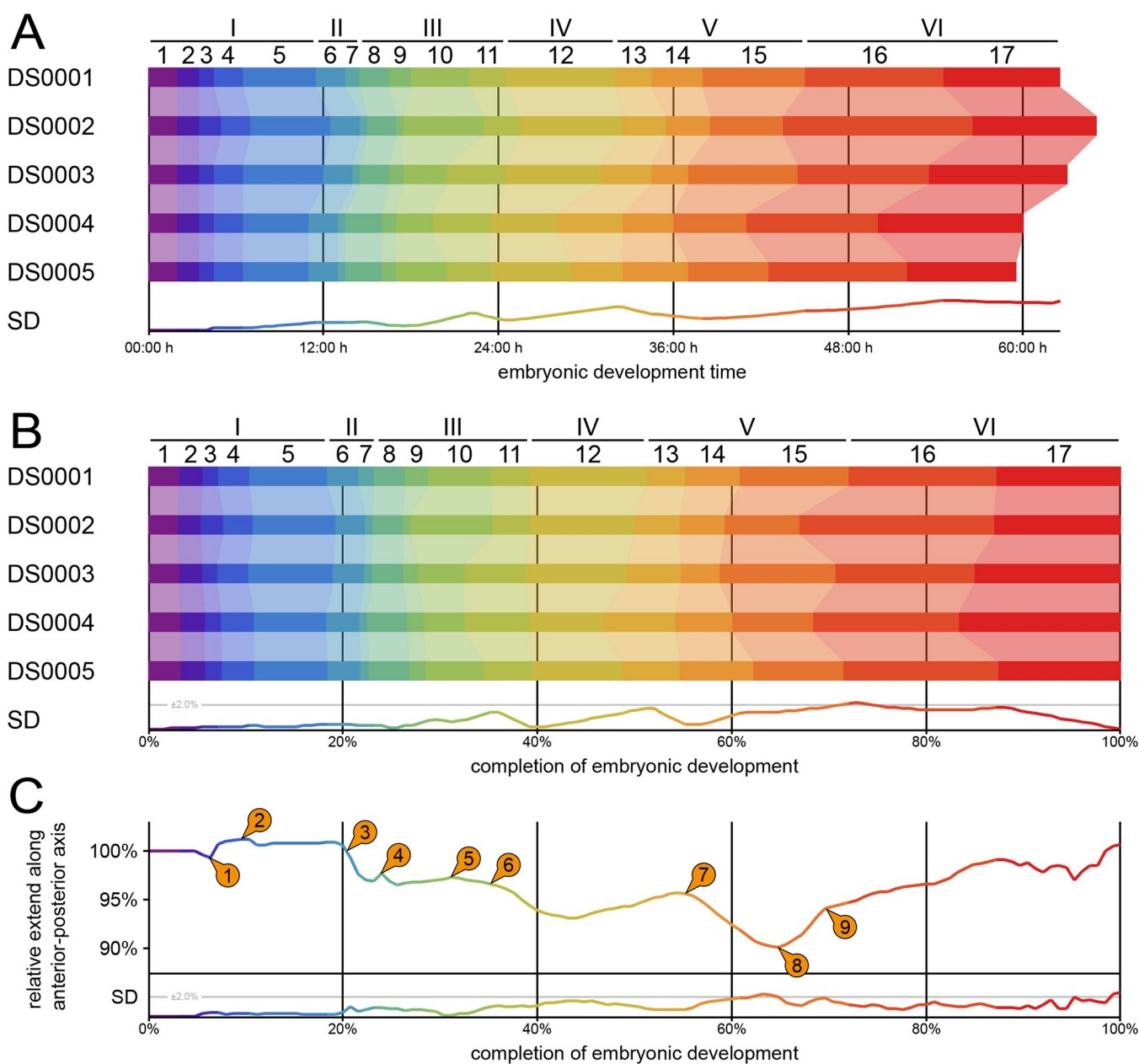

**Fig 1. Staging of medfly embryogenesis and morphological analysis of five datasets. (A)** Absolute stage duration stacked bar chart for DS0001–DS0005. **(B)** Normalized stage duration stacked bar chart for DS0001–DS0005. **(C)** Normalized embryo length graph with standard deviation for datasets DS0001–DS0003 as a function of embryonic development. The numbered arrows are referred to throughout the running text as well as in S2 and S3 Tables. SD, standard deviation.

- S2 Table provides chronological description of embryogenesis with the maximum temporal resolution. For longer-lasting processes, beginning and completion are indicated separately.

- S3 Table offers an overview of ten selected higher-level processes in chronological order. This glossary-like table indicates the embryogenetic events during the respective process and summarizes the morphogenic sequence in broad terms. Additionally, the processes are briefly embedded into an evolutionary context.

- S4 Table briefly describes all important structures and their precursors and/or successors in alphabetical order. This table also functions as a comprehensive glossary for the terminology–all structural terms written in italics throughout the publication are defined here.

- S1 Fig provides a stage-based quick lookup chart, in which each of the seventeen stages are outlined with a representative image, a temporal breakdown, and up to four bullet points highlighting the most important incidents per stage.

- S1 and S2 Videos illustrate embryonic morphogenesis in an intuitive dynamic fashion along the ventrolateral and dorsolateral axes or the ventral, lateral and dorsal axes, respectively.

## Summary

**(I) Blastoderm formation.** Blastoderm formation starts with egg fertilization and consists of five stages (1 to 5) outlined in S2 Table and Fig 2A and 2B. Stage 1 is characterized by pronuclear fusion, the resulting *zygote* is located in the inner regions of the *yolk* (Fig 2A, first column). During stage 2, the pre-blastoderm cleavage stage, the *zygote* undergoes multiple nuclear divisions. The resulting *zygotic nuclei* form a large syncytium within the inner regions of the *yolk*. After the initial nine synchronous divisions waves, the majority of *zygotic nuclei* start protruding towards the surface as part of the *peripheral migration* process. They can be reckoned at the end of stage 2 (Fig 2A, second column). Stage 3 begins when the first *zygotic nuclei* reach the surface (Fig 2A, third and fourth column, detail images ③④). While several nuclei give rise to the *pole buds* (Fig 2A, fourth column ②; Fig 2C), the large majority immediately start with the 10th and 11th synchronous nuclear division. When the last *zygotic nuclei* reach the surface and complete the 11th division, they turn into *blastoderm nuclei*. With the end of the *peripheral migration* process, the *zygote* turns into the *syncytial blastoderm*, which marks the beginning of stage 4. During this stage, the *syncytial blastoderm* undergoes two more parasynchronous nuclear divisions that run exclusively on the surface and proceed in a wave-like fashion from anterior to posterior (Fig 2B, first to fourth column).

As a consequence of the 10th and 11th synchronous nuclear division wave, the mean projected nuclei area drops to approximately one-third (Fig 2D) while the nuclei density increases approximately four-fold (Fig 2E). The nuclei are now distributed homogeneously on the surface. During stage 5, the *blastoderm nuclei* remain quiescent at their location for about four hours (Fig 2B, fifth column; Fig 3A, first and second column), which indicates that the *cellularization* process, which has previously been described [36], takes place during this period. At the end of stage 5, the medioventral *blastoderm nuclei* exhibit a wavy appearance (Fig 3A, first and second column, detail images), preceding the subsequent comprehensive morphogenic rearrangements during gastrulation.

During stage 2, the *yolk* begins withdrawing from the vitelline membrane at the anterior and posterior poles as part of the *repeated withdrawal* process (Fig 2C, first column) before reaching its relative maximum distance at the beginning of stage 3 (Fig 2A ②; Fig 2C, second column; cf. Fig 1C ). In parallel to the emergence of the *pole buds* at the posterior end of the *zygote*, the withdrawal begins to reverse (Fig 2C, third column). Complete reversal is accomplished at the end of stage 3 (Fig 2C, fourth column) and retained during stages 4 and 5 (Fig 2B).

The emergence of the *pole buds* at the posterior pole (Fig 2A, fourth column ②; Fig 2C) is part of the *germ cell dynamics* process that begins during stage 3 [37]. At the beginning of stage 4, those protuberances pinch off, turn into the *pole cells* (Fig 2B, first column), and become tightly grouped (Fig 2B, second to fifth column ). Previous investigations have discovered that the *pole cells*, in contrast to the *blastoderm nuclei*, are already separated by cell membranes

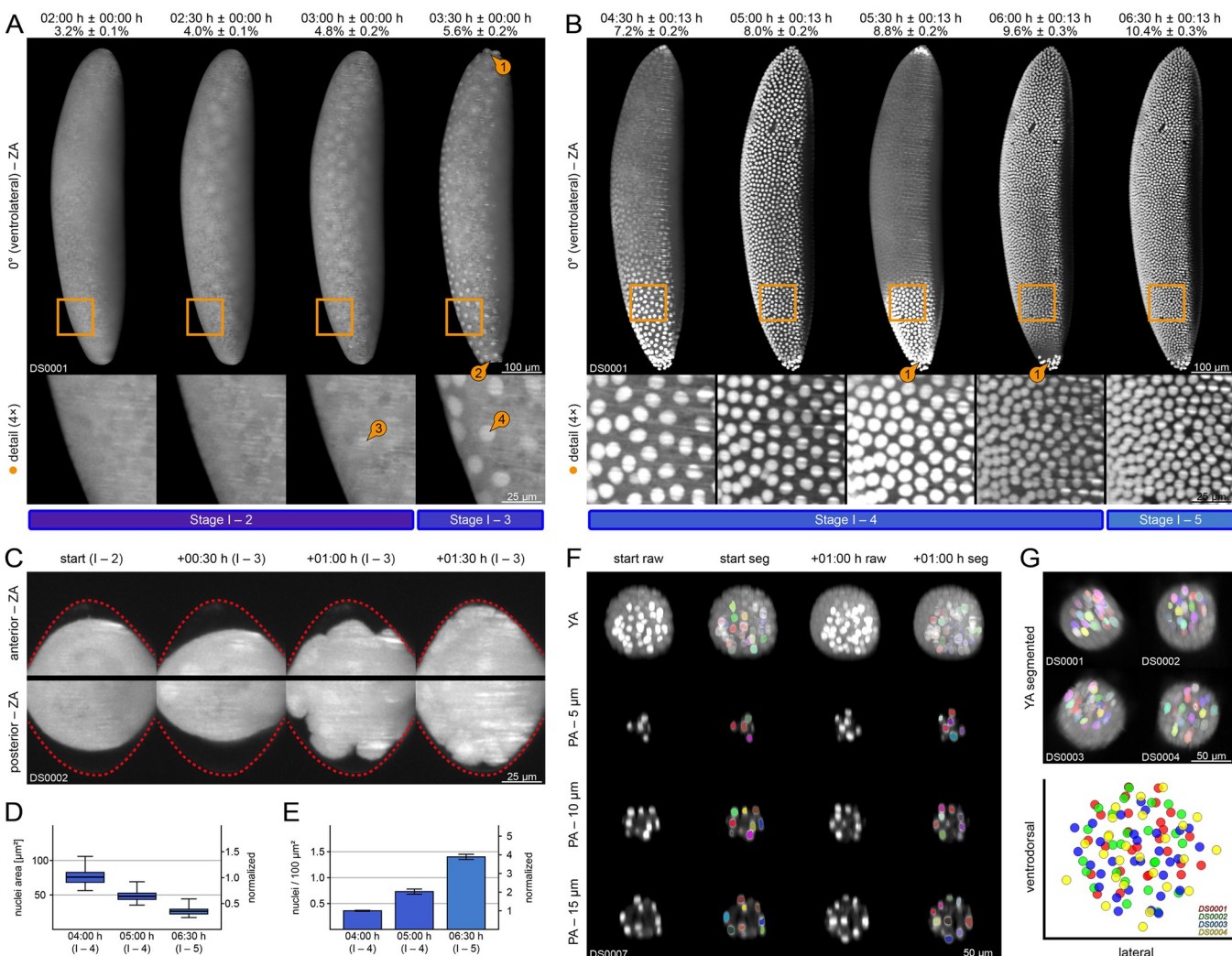

**Fig 2. Blastoderm formation (embryogenetic event I). (A)** Stages 2 and 3. Withdrawal of the *yolk* from the anterior and posterior ② poles, *zygotic nuclei* migration to the surface ③ and emergence of the *pole buds* at the posterior pole ②. **(B)** Stages 4 and 5. The 12th and 13th parasynchronous nuclear division of the superficially located *blastoderm nuclei*. The *pole buds* turn into the *pole cells* at the posterior pole and undergo an asynchronous mitotic division . **(C)** Withdrawal and reversal of the *yolk* from the vitelline membrane (red dashed line) at both poles and emergence of the *pole buds* at the posterior pole. **(D)** Projected area of the *zygotic nuclei* before the 12th, after the 12th and after the 13th parasynchronous nuclear divisions. **(E)** Projected density of the *zygotic nuclei* before, before the 12th, after the 12th and after the 13th parasynchronous nuclear divisions. **(F)** Segmentation of *pole cells* before and after their asynchronous division. **(G)** Segmentation and alignment of *pole cells* from four datasets. ZA, Z maximum projection with image adjustment; PA, single plane with image adjustment; seg, segmented.

[37]. Three-dimensional segmentation analyses (Fig 2F) indicate that the *pole cells* undergo asynchronous division, which results in a mean of 27.3 ± 2.6 identifiable nuclei at the end of stage 4. A conspicuous spatial organization pattern across DS0001–DS0004 was not found (Fig 2G). During stage 5, the *pole cells* remain quiescent.

**(II) Early gastrulation.** Gastrulation starts when the *blastoderm nuclei* begin rearranging after an extended period of quiescence and consist of two stages (6 and 7) that are summarized in S2 Table and Fig 3A–3C. At the end of the *cellularization* process, the *syncytial blastoderm* turns into the *cellular blastoderm*, which undergoes considerable morphogenic changes in a very short time period (Fig 3C–3E). The wavy appearance of the medioventral *blastoderm nuclei* intensifies (Fig 3A, first and second column), resulting in the *ventral furrow*, a

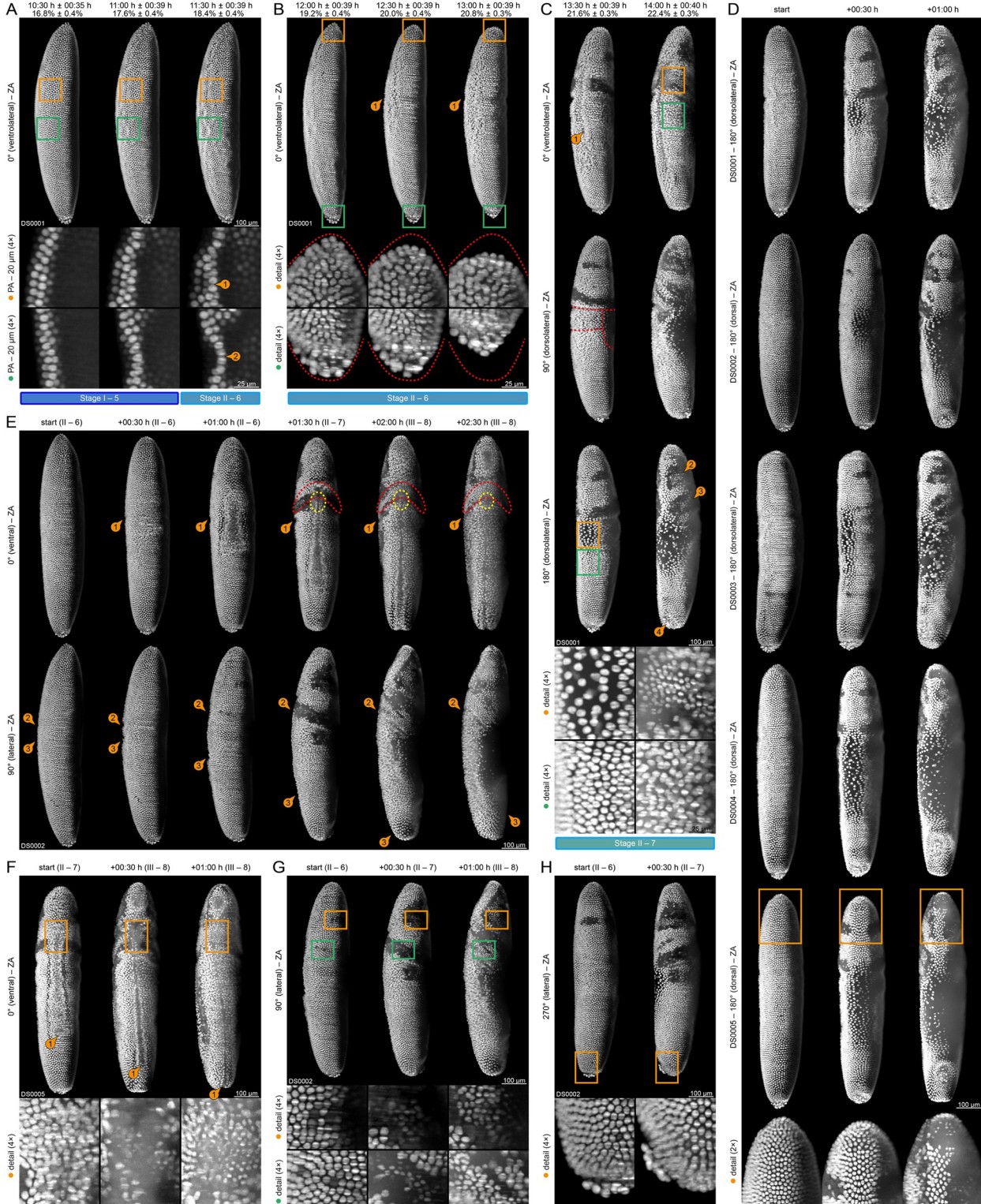

**Fig 3. Early gastrulation (embryogenetic event II). (A)** Stages 5 and 6. Wavy appearance of *blastoderm nuclei* and emergence of the anterior and posterior ② tips of the *ventral furrow*. **(B)** Stage 6. Withdrawal of the *head* and the *abdomen* from the vitelline membrane (red dashed line) at the anterior and posterior pole, respectively. Emergence of lateral indentions that precede the formation of the *cephalic furrow*. **(C)** Stage 7. Deepening of the *cephalic furrow*, extension of the *ventral furrow* and mitotic division of the *small and large lateral cell stripes* ②③. Differentiation of the *cellular blastoderm* into the *head*, the *thorax*, the *abdomen* and the *amnioserosa* (red dashed lines). Formation of the *gnathocephalon*. The *posterior*

*plate*, which carries the *pole cells*, turns into the *dorsal plate* ④. **(D)** Differentiation of the *amnioserosa* shown for five datasets. **(E)** Emergence of the *cephalic furrow* and *ventral furrow* ②③, emergence of the *anterior midgut primordium* (yellow dashed line) and formation of the *gnathocephalon* (red dashed line). **(F)** Emergence of the *anterior midgut primordium* and elongation of the *ventral furrow* . **(G)** Mitotic division of the *small and large lateral cell stripes*. **(H)** The *posterior plate*, which carries the *pole cells*, turns into the *dorsal plate*.

longitudinal cleft that arises medioventrally at the beginning of stage 6 (Fig 3A, third column ②; Fig 3B, 3C ; 3E ②③). The *ventral furrow* is involved in the *germ layer specification* process, specifically mesoderm internalization, which occurs during the next few hours. While its anterior tip remains nearly stationary (Fig 3E ②; 3F), the posterior tip extends posteriad and curls around the posterior pole at the transition from stage 7 to stage 8 (Fig 3E ③; 3F ). During this process, additional blastodermal cells along the ventral midline are recruited and incorporated. The initial groove width of the *ventral furrow* spans about five to seven cells (Fig 3E, third column), while the width of the residual transient gap, which remains upon invagination, spans only about one to two cells (Fig 3E, fifth column). Throughout continuous closure, the *ventral furrow* differentiates into the internal *mesodermal layer*, which is widely covered by a superficial *ectodermal layer*. A minor part around the anterior tip (Fig 3E, yellow dashed line; Fig 3F) gives rise to the *anterior midgut primordium*, which is of endodermal origin. At the transition from stage 7 to stage 8, the *anterior midgut primordium* internalizes and undergoes mitotic division (Fig 3F, detailed images).

At the end of stage 6, the *cellular blastoderm* withdraws from the anterior and posterior poles (Fig 3B, detail images; cf. Fig 1C ③). In parallel, the *posterior plate*, to which the *pole cells* adhere, emerges at the flattened posterior tip of the *cellular blastoderm*. At the end of stage 7, the *posterior plate* shifts its position from the posterior pole towards the dorsal side and turns into the *dorsal plate* as part of the *germ cell dynamics* process (Fig 3C, second column; Fig 3H).

Shortly after the *ventral furrow* becomes apparent, symmetrically arranged lateral indentions arise anteromedially, which precede the formation of the *cephalic furrow* (Fig 3B ; 3E, second column ), leading to a radial alignment of the nuclei in the triangular area between the indentions and the *ventral furrow* at the end of stage 6 (Fig 3E, third column). At the beginning of stage 7, the indentions expand, with moderate anterior inclination, towards the ventral midline and approach the anterior tip of the *ventral furrow* (Fig 3E, fifth and sixth column). This results in a deepening of the *cephalic furrow*, which morphologically separates the *cellular blastoderm* into the anterior *head* region the medio-ventral *thorax* region (Fig 3C). The *cellular blastoderm* differentiates furthermore into the posterior *abdomen* and the mediodorsal *amnioserosa* (Fig 3C, red dashed lines). The *head*, the *thorax* and the *abdomen* constitute the *embryo*–the entirety of tissue regions that become part of the larva after hatching–in contrast to the extra-embryonic *amnioserosa* and the *yolk sac*, which become degraded later. The *head* withdrawal soon reaches a relative maximum and begins reversing (Fig 3C, second column; cf. Fig 1C).

As part of the *extra-embryonic membrane folding* process, the *amnioserosa* extends moderately laterally and strongly posteriad during the next hour (Fig 3C, first column, detail images; Fig 3C, second column; Fig 3D). The cells of the *amnioserosa* do not proliferate, which leads to a sparse distribution of the nuclei over the whole area of the extra-embryonic membrane at the end of stage 7 (Fig 3D, second and third column). Surprisingly, the medfly does not develop an anterior and posterior transverse fold.

Early gastrulation is also characterized by mitotic divisions of four cell clusters within the *head*. At the beginning of stage 7, the *small and large lateral stripes* undergo mitotic division (Fig 3C, first column; Fig 3G, detail images). After both clusters proceed through the telophase (Fig 3C, second column ②③; Fig 3G, detail images), the cells around those cluster also begin

with mitotic division. After completion at the end of stage 7, the resulting nuclei are considerably smaller (Fig 3C, second column, detail images; Fig 3G, detail images) and the *head* has remodeled–while the descendants of the large lateral stripe and its surrounding cells, which are located directly anterior of the *cephalic furrow*, form the postero-ventral *gnathocephalon* (Fig 3C, second column; cf. Fig 4A, first column ②), the remaining cells constitute the antero-ventral / dorsal *procephalon*. Together with the *thorax* and the *abdomen*, the *gnathocephalon* composes the *germband*.

**(III) Germband elongation.**   Germband elongation starts when the posterior tip of the *germband* begins anteriad elongation along the dorsal side with a high migration velocity (relative to subsequent stages) and consists of four stages (8 to 11) that are summarized in S2 Table and Fig 4A and 4B. At the beginning of stage 8, the posterior tip of the *germband* curls around the posterior pole and accomplishes the first 400 µm of anteriad migration within about two hours (Fig 4A and 4C). The migration velocity decelerates two times, once at the beginning of stage 9 and once at the beginning of stage 10, so that the final 100 µm are accomplished within about five and a half hours. Once the *germband* is fully elongated, the *amnioserosa* has folded completely and the *yolk sac* has migrated from the surface to the inner regions (Fig 4B, third and fourth column). During stage 11, the posterior tip of the *germband* remains in a quiescent position (Fig 4D and 4E) while the *gnathal protuberances* emerge from the *gnathocephalon* as part of the *metamerization* process (Fig 4B, detail images ②③).

Shortly after the beginning of stage 8, the *head* withdrawal from the vitelline membrane at the anterior pole reverses completely (Fig 4A, second column; cf. Fig 1C ④), but begins withdrawing again during stage 10 (Fig 4B, second column; cf. Fig 1C ⑤) before reaching a momentary standstill at the beginning stage 11 (Fig 4B, third column; cf. Fig 1C ⑥). The *abdomen* withdrawal from the vitelline membrane at the posterior pole reaches a momentary standstill at the end of stage 8 (Fig 4A, second and third column; cf. Fig 1C) and continues withdrawing at the beginning of stage 11 (cf. Fig 1C ⑥).

Around the same time as the *germband* begins anteriad elongation, the *stomodeal cell plate* emerges antero-ventrally from the *procephalon* (Fig 4A, first column ; 4F, first column). Shortly afterwards, the *stomodeal cell plate* sinks inwards, turns into the *stomodeal invagination* (Fig 4A, second column, ②), and connects with the *anterior midgut primordium* (Fig 4F, second column ). At the beginning of stage 9, the *stomodeal invagination* turns into the *stomodeum* (Fig 4A, third and fourth column ③; Fig 4F, fifth column), which becomes the antero-medial part of the *digestive tract* as part of the *digestive system formation* process. In parallel, the *cephalic furrow* levels out (Fig 4A, third column; Fig 4G, first row of detail images).

During early elongation, the posterior tip of the *germband* carries the *dorsal plate*, to which the *pole cells* adhere (Fig 4A, first column; Fig 4C, detail images ; Fig 4H, first to third column). The *dorsal plate* turns into the *posterior midgut primordium* during stage 8 (Fig 4A, second column ⑤), which subsequently invaginates, but still carries the *pole cells* (Fig 4C, detail images ②). The *germ layer specification* process ends since all components are now allocated to their respective layers. Shortly afterwards, at the posterior tip of the *ventral furrow*, the *ectodermal layer* gives rise to the *amnioproctodeal invagination*, a tubular structure that surrounds the *posterior midgut primordium* (Fig 4A, second column; Fig 4C, second column). As part of the *digestive system formation* process, *amnioproctodeal invagination* differentiates into the *proctodeum* (Fig 4A; cf. Fig 4, detail images, yellow dashed line) and the *proctodeal opening* (Fig 4A, third column; cf. Fig 4C, detail images, red dashed line) at the beginning of stage 9. The *proctodeal opening* changes from a dorsal to an anterior orientation at the end of stage 9. Thus the *pole cells* vanish from sight (Fig 4A, fourth column ⑥; Fig 4H, fourth column) and the *germ cell dynamics* process concludes.

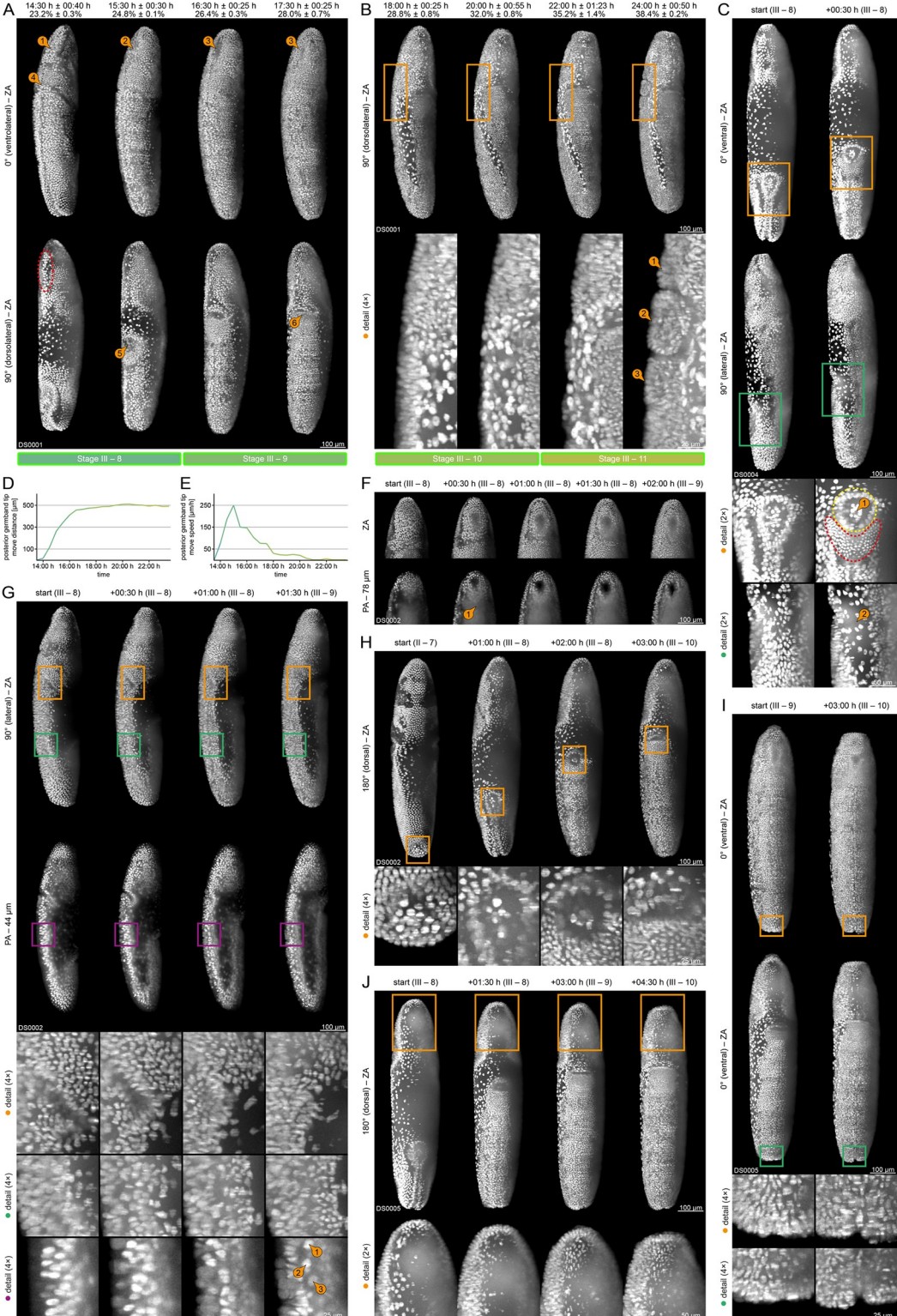

**Fig 4. Germband elongation (embryogenetic event III). (A)** Stages 8 and 9. Anteriad elongation of the posterior tip of the *germband*. Emergence of the *stomodeal plate* ①, which turns into the *stomodeal invagination* ② and later into the *stomodeum* ③. Internalization of the *anterior midgut primordium* ④. The dorsal plate turns into the *posterior midgut primordium*. Emergence of the *optic lobe primordium* (red dashed line) and the *amnioproctodeal invagination* ⑤, which differentiates into the *proctodeum* and the *proctodeal opening* ⑥. **(B)** Stages 10 and 11. Anteriad elongation and quiescence of the

posterior tip of the *germband*. Emergence of the *gnathal protuberances* from the *gnathocephalon*, white arrows indicate the respective gnathal segment. **(C)** Emergence of the *amnioproctodeal invagination* (yellow and red dashed lines) around the *posterior midgut primordium* ②, which carries the *pole cells* . **(D)** Migration distance of the posterior tip of the *germband* over a six-hour period. **(E)** Migration speed of the posterior tip of the *germband* over the same six hour period. **(F)** Emergence of the *stomodeal cell plate*, which turns to the *stomodeal invagination*. The *stomodeal invagination* connects to the *anterior midgut rudiment* and turns into the *stomodeum*. **(G)** Leveling of the *cephalic furrow*, differentiation of the *ventral epidermal primordium* and the *lateral epidermal primordium* and segregation of *neuroblasts* ②③. **(H)** Invagination of the *posterior midgut primordium* as well as differentiation and internalization of the *proctodeum*. **(I)** Emergence of the *ventral mesectoderm*. **(J)** Mitotic division in the *optic lobe primordium*.

At the end of stage 8, the *ectodermal layer* differentiates into the *ventral epidermal primordium*, which composes of cells with large nuclei and the *lateral epidermal primordia*, which composes of cells with small nuclei (Fig 4A, first and second column; Fig 4G, detail images). Germband elongation is also the embryogenetic event where neurogenesis begins: Firstly, after the emergence of the *stomodeal cell plate* at the beginning of stage 8, all remaining ventrally and laterally located cells of the *procephalon* undergo mitotic division, sparing a longitudinal stripe of cells with a scraggy outline–the *optic lobe primordium* (Fig 3D, detail images; Fig 4A, first column, red dashed line). The primordium is a shallow groove with a width of around three to four cells and extends along the dorsal midline from the anterior tip of the *procephalon* to the anterior rim of the *amnioserosa*. During stage 8, the *optic lobe primordium* begins multiple cycles of mitotic division that last until the end of stage 10 (Fig 4J). Secondly, in the *thorax* and the *abdomen*, the first *neuroblasts* segregate from the *ventral epidermal primordium* at the beginning of stage 9 (Fig 4G ②③). Thirdly, the *ventral furrow* levels out at the beginning of stage 9 (Fig 4A, third column; Fig 4I, first column). About two hours later, the *ventral mesectoderm* emerges as a longitudinal stripe of mesectodermal cells from the *ventral epidermal primordium* at the same position (Fig 4I, second column). Any further neurodevelopmental processes cannot be properly identified, and thus, neurogenesis is not described further within this study (S5 Table).

**(IV) Germband retraction.** Germband retraction starts when the posterior tip of the *germband* begins posteriad retraction along the dorsal side and consists of one stage (12) that is summarized in S2 Table and Fig 5A. In contrast to germband elongation, which had three periods with high, moderate, and low migration velocities, the posterior tip of the *germband* retracts with a relatively constant velocity over the next seven and a half hours (Fig 5A, 5B). The retraction is directly linked to the *extra-embryonic membrane folding* process–the *amnioserosa* unfolds for a second and the underlying *yolk sac* protrudes towards the medio-dorsal surface (Fig 5A, second to fourth column). At the end of stage 12, when the retraction process in completed, the *proctodeal opening* is located posterior-dorsally (Fig 5A, fourth column ).

During germband retraction, distinct activities associated with the *repeated withdrawal* process take place. Three hours after the beginning of stage 12, the *abdomen* withdrawal reaches a relative maximum and begins reversing (Fig 5A, second and third column ②; cf. Fig 1C), while the *head* withdrawal remains at a standstill during the whole period of germband retraction (Fig 5; cf. Fig 1C). During this state, two hours after *germband* retraction begins, the *clypeolabrum* emerges antero-dorsally from the *procephalon* (Fig 5A, detail images). Over the next few hours, the *clypeolabrum* becomes thinner and more pronounced, but retains its antero-dorsal orientation (Fig 5B, detail images).

During stage 12, the *dorsal epidermal primordia* emerge in an anterior-to-posterior fashion from the *lateral epidermal primordia* (Fig 5B, second to fourth column ). Shortly after the *abdomen* withdrawal has reached a relative maximum as part of the *repeated withdrawal* process and begins reversing, the *intersegmental furrows* arise and separate the *thorax* into three (Fig 5B ②③④) and the *abdomen* into nine segments. Together with the emergence of the

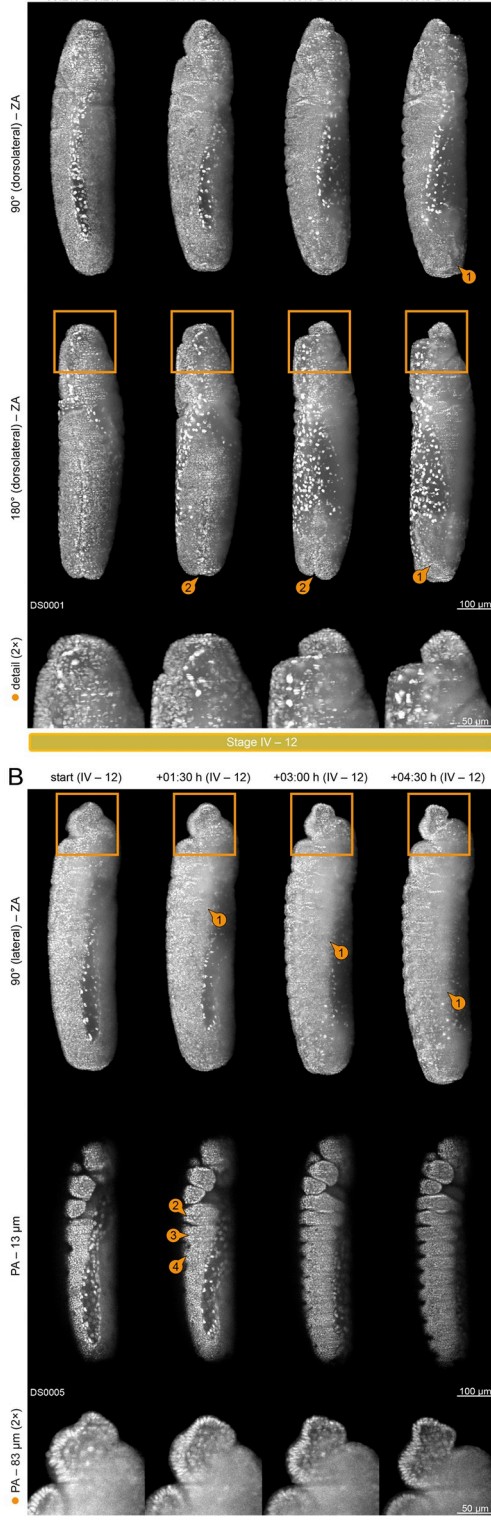

**Fig 5. Germband retraction (embryogenetic event IV). (A)** Posteriad retraction of the posterior tip of the germband and posterior-dorsal localization of the *proctodeal opening* . Emergence of the *clypeolabrum* and advent of the *intersegmental grooves*. Retraction and reversal of the abdomen from the vitelline membrane at the posterior pole ②. **(B)** Emergence of the *clypeolabrum* and the *dorsal epidermal primordia* . Advent of the *intersegmental grooves* in the *thorax* ②③④ and *abdomen*.

*gnathal protuberances* in the *gnathocephalon* during stage 11, the *germband* is now divided into fifteen segments as part of the *metamerization* process.

**(V) Dorsal closure.** Dorsal closure starts when the posterior tip of the *germband* completes posteriad retraction and consists of three stages (13 to 15), summarized in S2 Table and Fig 6A and 6B. In the first morphological change of stage 13, the *anterior dorsal gap*, a shallow transversal groove, arises between the *procephalon* and the *amnioserosa* (Fig 6A, second column ; Fig 6C, first to third column). In parallel, the *gnathal protuberances* give rise to the bilateral *dorsal folds* (Fig 6D, first column) and subsequently differentiate into the *mandibular buds* (Fig 6A, second column ②; Fig 6C, first column ; Fig 6E, first column ; Fig 6F, third column), the *maxillary buds* (Fig 6A, second column) and the *labial buds* (Fig 6A, second column; Fig 6C, second column ②). The *maxillary buds* fuse with the antero-ventral region of the *procephalon* to form the *antennomaxillary complexes* (Fig 6A, fifth column ⑤; Fig 6C, fourth column ③; Fig 6E, third and fourth column ④; Fig 6F, fourth column) while the *labial buds* migrate ventrad and eventually fuse (Fig 6A, fifth column ④; Fig 6C, second to fourth column ②; Fig 6E, second to fourth column ②; Fig 6F). The *anterior dorsal gap* is only transient, and it levels out during the next few hours (Fig 6A, third and fourth column ; Fig 6C, fourth column ④).

The *head involution* process begins as the *ventral epidermal primordium*, and the *lateral epidermal primordia* (Fig 6A, first column, red dashed lines) begin with anteriad migration to form an envelope around the ventrolateral part of the *head*. Involution of the dorsal part is slightly delayed and does not involve the *dorsal epidermal primordia*. Instead, the *dorsal folds* fuse along the *anterior dorsal gap* and begin with anteriad migration, covering the *procephalon* and the *clypeolabrum* gradually (Fig 6D). The collective anteriad migration of the epidermal primordia and the *dorsal fold* is accompanied by a co-motion of the bilaterally located *antennomaxillary complexes* (Fig 6E, third and fourth column ④; Fig 6F; S3 Video) and a posteriad counter-motion of the *clypeolabrum* (Fig 6A, fifth column; Fig 6B and 6C, fourth column; Fig 6E, third and fourth column; Fig 6F, third and fourth column), the *mandibular buds* (Fig 6A, fifth column; Fig 6B and 6C, fourth column ; Fig 6E, third and fourth column ; Fig 6F, third and fourth column) and later on also the fused *labial buds* (Fig 6B and 6E, third and fourth column ②; Fig 6F, third and fourth column), which retract into the inner regions of the *embryo* (S3 Video).

During dorsal closure, the posterior tip of the germband flips into the posterior pole (Fig 6B), and the *intersegmental grooves* of the *thorax* and the *abdomen* transiently level out (Fig 6B, third column). Anteriorly within the *thorax*, the *anterior midgut primordium* and the *posterior midgut primordium* fuse to form the *midgut* (Fig 6A, third column ③; Fig 6G, third column, yellow dashed line), which internalizes the *amnioserosa* and the *yolk sac* at the end of the *extra-embryonic membrane folding* process. The internalization procedure requires the *amnioserosa* to fold for a second time, which seems to be partially mediated by the dorsad migration of both *dorsal epidermal primordia*. This approaching motion marks the beginning of the *dorsal zippering* process–throughout the next several hours, the primordia establish antero- and posteromedially contact and fuse continuously to form a tight seam along the dorsal midline. The anterior leading edge migrates slowly (Fig 6G, first to third column ), partially covering the *midgut* (Fig 6G, third column, yellow dashed line). In contrast, the posterior leading edge migrates slightly faster (Fig 6G, first to third column ) and covers the *proctodeum* completely (Fig 6G, first column, red dashed line). After completion of anterior and posterior fusion at the beginning of stage 15, the anterior and posterior leading edges continue with median fusion (Fig 6G, fourth and fifth column ③④).

At the beginning of stage 14, withdrawal of the *head* continues after the momentary standstill (Fig 6A; cf. Fig 1C ⑦). Both the *head* and the *abdomen* reach their absolute withdrawal

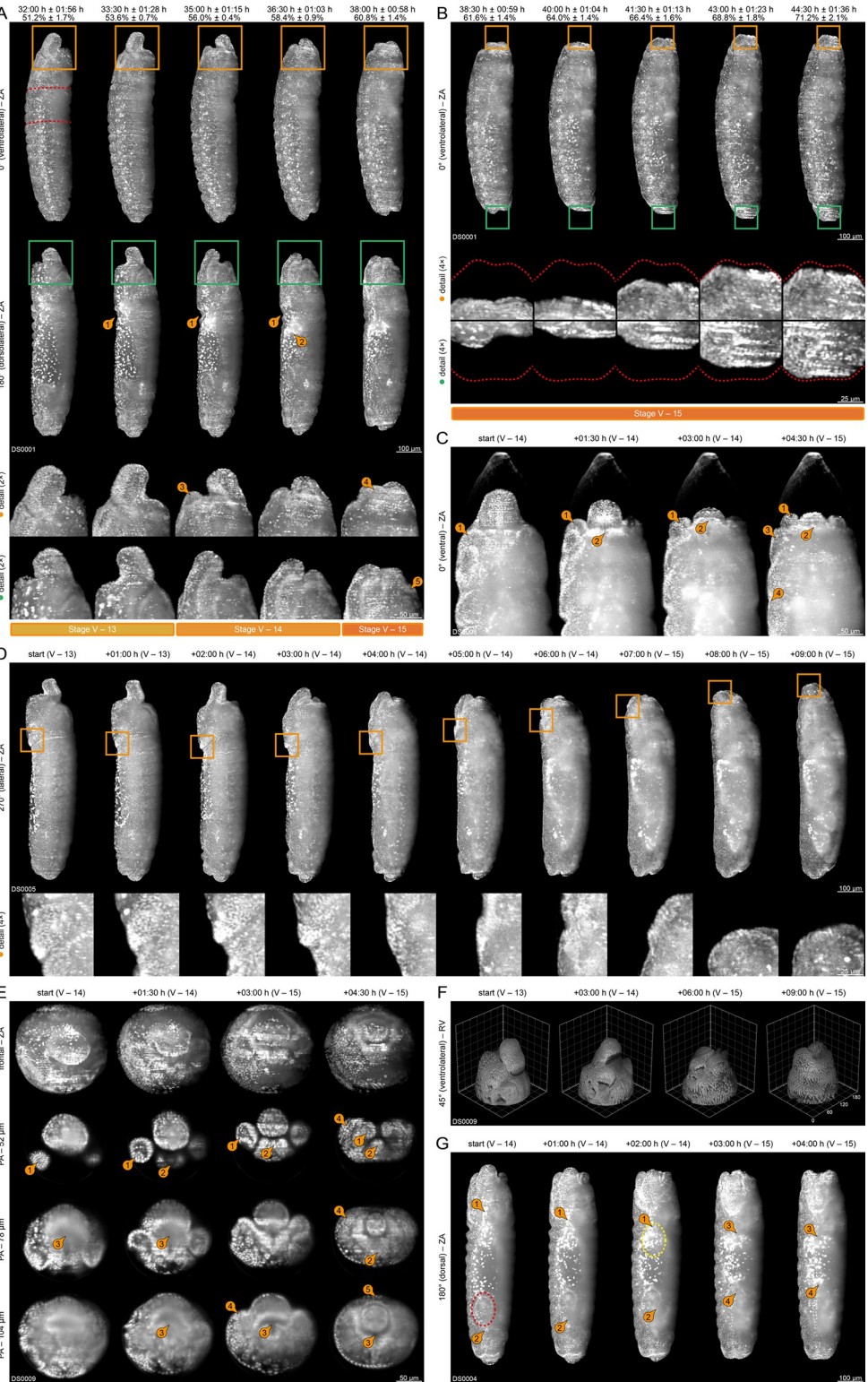

**Fig 6. Dorsal closure (embryogenetic event V). (A)** Stages 13, 14 and 15. Anteriad migration of the *ventral epidermal primordium* and the *lateral epidermal primordia* (red dashed lines) as well as dorsolaterad migration and fusion of the *dorsal epidermal primordia*. Advent of the *anterior dorsal gap* and formation of the *midgut* ③. Morphogenic change of the *clypeolabrum*. Differentiation of the *mandibular buds* ③, the *maxillary buds* and the *labial buds*. Fusion of the *labial buds* ④. Emergence of the *antennomaxillary complex* ⑤. **(B)** Stage 15. Withdrawal of the *head* and the *abdomen*

from the vitelline membrane at the anterior and posterior pole, respectively, and transient leveling of the *intersegmental grooves*. Retraction of the *clypeolabrum*, the *mandibular buds* and the fused *labial buds*. **(C)** Retraction of the *clypeolabrum*, the *mandibular buds* and the fused *labial buds* ②. Emergence of the *antennomaxillary complex* ③. Leveling of the *anterior dorsal gap*. **(D)** Anteriad migration of the fused *dorsal folds*. **(E)** Retraction of the *clypeolabrum*, the *mandibular buds*, the fused *labial buds* ② and the *stomodeum* ③ as well as emergence of the *antennomaxillary complex* ③. Anteriad migration of the *dorsal fold* ⑤. **(F)** Volume rendering of head involution. **(G)** Anterior and posterior ② as well as median ③④ fusion of the *dorsal epidermal primordia*, which cover the *proctodeum* (red dashed line) and the *midgut* (yellow dashed line).

maxima during stage 15 (Figs 1CB and 6B ⑧) so that the relative longitudinal extent of the *embryo* has dropped to nearly 90% of its initial value. Withdrawal reverses relatively fast within the next three hours (Fig 1C ⑨).

**(VI) Muscular movement.** Muscular movement starts when the *dorsal epidermal primordia* complete median fusion and turn into the *dorsal epidermis*, thus ending the *dorsal zippering* process and consists of two stages (16 and 17) that are summarized in S2 Table and Fig 7A. One of the most remarkable occurrences of this embryogenetic event is the onset of regular muscular movement within the *embryo* (Fig 7B). At the beginning of stage 16, both the *ventral epidermal primordium* and the *lateral epidermal primordia* complete their anteriad migration and turn into the *ventral epidermis* and *lateral epidermis*, respectively (Fig 7A, first column, red dashed lines; cf. Fig 6A, first column, red dashed lines). Slightly later, the fused *dorsal folds* also complete anteriad migration and turn into a pocket-like structure, the *dorsal pouch* (Fig 7A and 7C, first row of detail images, red dashed lines), which ends the *head involution* process. Of all head appendages, only the anterior, fused part of the *antennomaxillary complexes* is still superficial, while the posterior part, the *clypeolabrum*, the *mandibular buds* and the *labial buds* contribute to form the *atrium*. The *digestive system formation* process ends as the anterior *atrium*, the anteromedial *stomodeum*, the medial *midgut*, the posteromedial *proctodeum* and the posterior *proctodeal opening* consecutively connect to form the *digestive tract*.

At the beginning of stage 17, the withdrawal of the *head* and the *abdomen* from the vitelline membrane nearly completely reverses (Fig 7A; cf. Fig 1C). During muscular movement, the posterior tip of the *ventral cord* shortens gradually over a period of several hours until it reaches its final position between the 4$^{th}$ and 5$^{th}$ segment of the *abdomen* (Fig 7C, second row of detail images). Ultimately, the *embryo* hatches from the egg and turns into the larva (Fig 7D)–the completion of embryonic development and the beginning of the second phase of the life cycle.

## Discussion

In this study, we characterize the entire embryonic morphogenesis of the medfly qualitatively and quantitatively, providing an integrative framework for future studies. Since the medfly is primarily used as a model organism in agriculture-related research due to its status as a pest species, it has only been occasionally used in developmental biology. Our study shows that this species is a well-suited model organism for developmental biology and a promising candidate to strengthen the comparative approach.

### Our approach in comparison to traditional strategies

In contrast to similar studies that have analyzed the embryonic development of other insect species and established staging systems (twenty-one respective publications are summarized in S1 Table), our study is based on long-term fluorescence live imaging data. These data were acquired by imaging embryos from a stable homozygous transgenic line that ubiquitously and constitutively expresses nuclear-localized EGFP using LSFM. Due to its intrinsic

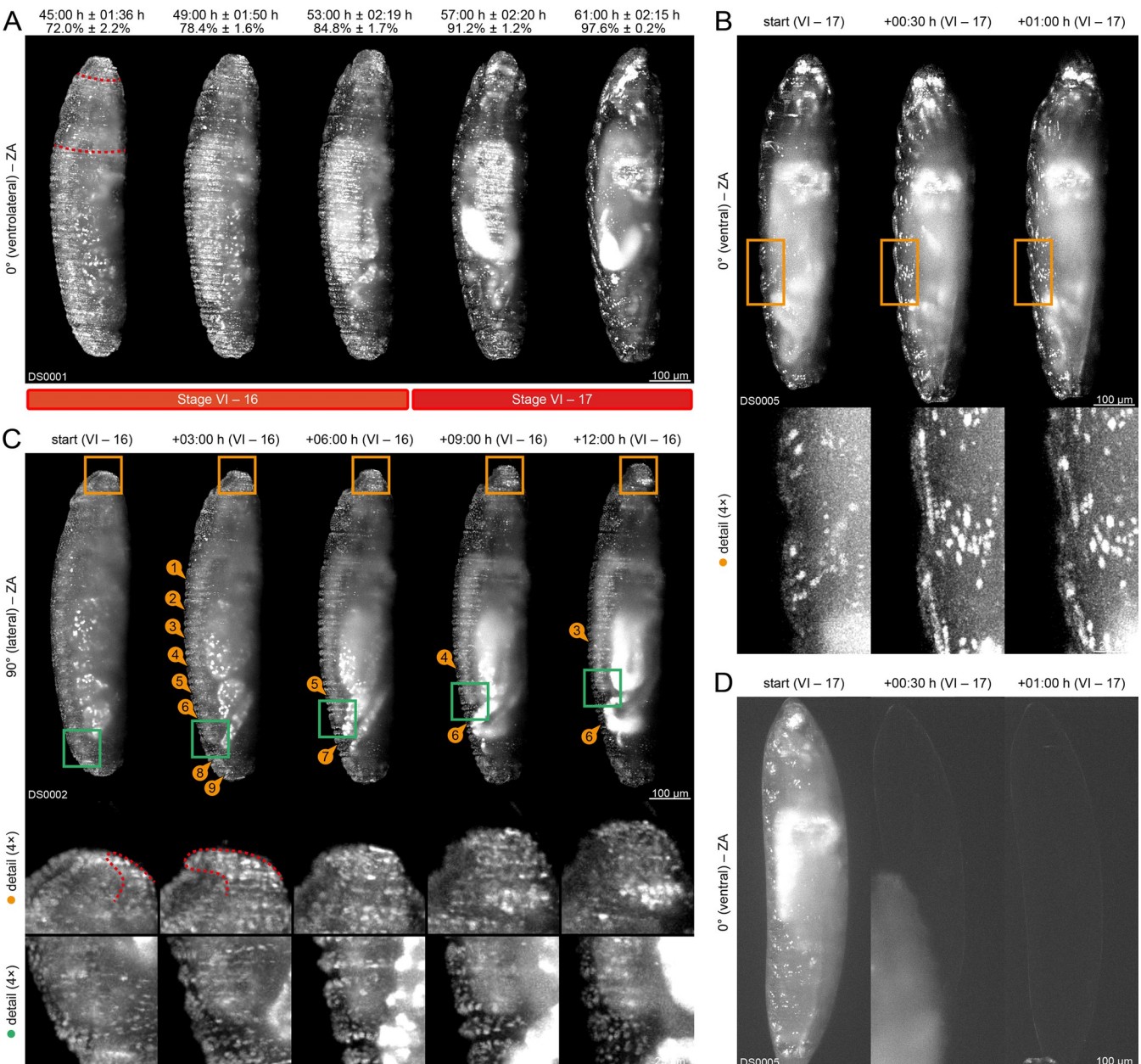

**Fig 7. Muscular movement (embryogenetic event VI). (A)** After medial fusion, the *dorsal epidermal primordia* turn into the *dorsal epidermis*. Re-emergence of the *intersegmental grooves* and formation of the *atrium*. **(B)** Onset of regular muscular movement. **(C)** After migration, the fused *dorsal folds* turn into the *dorsal pouch*. Shortening of the *ventral cord*, white arrows indicate the respective abdominal segment. **(D)** Hatching of the larva.

characteristics, *e.g.*, low photo-bleaching, nearly no photo-toxicity, intrinsic optical sectioning, very good signal-to-noise ratio, and the opportunity to image along multiple directions, LSFM is a well-suited technique for descriptive and analytical studies of insect embryogenesis.

In the past, the methods of choice have mainly been fixation and staining with non-fluorescent and fluorescent dyes, mechanical sectioning, or scanning electron microscopy. In those approaches, only one 'snapshot' of a specimen can be acquired. Therefore, to describe embryogenesis as a general process, many specimens are required, resulting in a high amount of experimental work and, even more crucial, certain statistical errors. For example, it is

cumbersome and imprecise to calculate temporal standard deviations by using fixed and stained samples. Also in traditional approaches, quality control can be challenging. Our quality criterion is the healthy development of the imaged specimen–out of nine imaged embryos, eight (DS0001–DS0005, DS0007–DS0009) developed into fully functional adults, and data derived from the improperly developing individual was disregarded.

However, the fluorescence live imaging approach also suffers from several limitations. Firstly, respective transgenic lines are a prerequisite for long-term fluorescence live imaging, and the descriptive scope depends on the available fluorescent labels. In our study, for example, the cellularization process at the end of blastoderm formation (stage I–5) is only known from previous studies [36, 37] as a membrane-labeled transgenic line is not available yet (cf. S5 Table). Secondly, the available image data are limited regarding, *e.g.*, spatial resolution. To investigate smaller structures, super-resolution approaches [52] or electron microscopy [53] need to be used. Thirdly, although the optical section capabilities of LSFM is already quite good, structures and processes within the inner regions of the embryo could not be reliably identified due to the relatively large size of the embryo (cf. S5 Table). To characterize meso-derm- and endoderm-derived structures, the available live imaging data may be comple-mented, *e.g.*, with data from fixed specimens that have either been mechanically sectioned or subjected to optical clearing [54] or volume expansion procedures [55].

## Quantitative analyses

In traditional approaches to studying embryonic development, structures and processes are often described only qualitatively (cf. S1 Table). However, extensive quantitative information is available for well-established model organisms such as the fruit fly. The lack of quantitative data in other systems is due to the fact that extracting such information from data obtained using classic methods is often labor-intensive, complicated, prone to error, or even impossible. One of the major advantages of our fluorescence live imaging approach is the overall high quality of the data, which facilitates the straightforward and efficient extraction of a wide range of quantitative metrics. This can enhance our multi-scale understanding of developmental processes, *e.g.*, at the intersection of gene regulatory networks and morphogenesis. For instance, the standard deviation curve of the relative completion of embryonic development reveals three distinct peaks, *i.e.*, (i) at the transition of germband elongation to germband retraction, (ii) at the transition of germband retraction to dorsal closure, and (iii) at the transi-tion from dorsal closure to muscular movement (cf. Fig 1B). Notably, similar peaks do not appear during the embryogenic events themselves, indicating that they likely arise from large-scale changes in gene expression patterns that coordinate these radical morphogenic realign-ments. It is reasonable to assume that developmental cascades of this magnitude are prone to a certain temporal variability.

In this study, we present several examples of quantitative data extraction, but we anticipate that, depending on the specific scientific question being addressed, even more quantitative data can be extracted and that these analytical processes can also be automated. Moreover, detailed quantifications of how, *e.g.*, mutations impact development at different stages could augment the design of more precise control strategies such as the precision guided sterile insect technique (pgSIT) [56] or dedicated gene drive systems, which require stage-specific promoters and time-sensitive expression of transgenes for optimal functionality.

## Developmental differences between the medfly and other insect species

In the evolutionary context, one of the most interesting insights of this study concerns the for-mation of extra-embryonic membranes. Similar to the fruit fly *Drosophila melanogaster* but

unlike more distantly related species, *e.g.*, the scuttle fly *Megaselia abdita* and the moth midge *Clogmia albipunctata*, the medfly gives rise to only the dorsally located amnioserosa. However, the relative timing between amnioserosa differentiation and germband elongation differs considerably between these two species. In the medfly, the amnioserosa differentiates already during gastrulation as a layer of widely spaced cells on the dorsal side between the posterior plate and the head during stage 7 (Fig 8A, upper row, first to fourth column). During germband elongation, as the germband tip migrates anterior, the amnioserosa drastically changes shape–it becomes dorsally compressed and expands in the form of 'lateral horns' over the flanks of the embryo until the end of stage 10 (Fig 8A, upper row, fifth to eight column). This morphogenic cascade is similar to the closely related Queensland fruit fly *Bactrocera tryoni* [57] but stands in strong contrast to the fruit fly, where the primordial amnioserosa cells remain clustered during gastrulation (Fig 8A, lower row, first to third column), and the subsequent lateral expansion process, which starts at the transition from stage 8 to stage 9, occurs without dorsal compression (Fig 8A, lower row, fourth to eight column). We hypothesize that a 'premature' morphogenic differentiation of the amnioserosa during gastrulation, as observed in this study for the medfly, may represent an apomorphic trait since other, more basally located dipteran species such as the scuttle fly [12, 14–16] and the moth midge [11] also appear to form their extra-embryonic membranes after early gastrulation. Extra-embryonic membranes in holo- and hemimetabolous insects are currently ambitiously investigated across various insect model organisms [58, 59], including non-dipteran species such as the red flour beetle *Tribolium castaneum* [60–65] and the milkweed bug *Oncopeltus fasciatus* [66–68]. Our description of the amnioserosa dynamics in the medfly thus contributes valuable insights to the understanding of the structure, function and evolutionary history of extra-embryonic membranes.

Another notable difference involves the formation of epithelial folds during early embryogenesis. For example the anterior and posterior transverse folds–characteristic transient structures in the fruit fly [38] as well as in the scuttle fly [70] during gastrulation and early germband elongation–are absent in both the medfly and the Queensland fruit fly [57]. In the fruit fly, only low levels of myosin and cortical actin are observed in the dorsal epithelium at the beginning of germband elongation [71], with no dedicated myosin accumulation prior to dorsal fold formation [72]. These findings led to the theory that the migrating germband contributes considerably to the formation of these folds by compressing the dorsal epithelium [73]. If this is correct, the absence of posterior folds in the medfly and Queensland fruit fly could relate to their egg shape, specifically the ratios of the anterior-posterior axes to the respective diameters, which is approximately twice as large in these two Tephritid species compared to the fruit fly and the scuttle fly. Notable, however, the Queensland fruit fly develops four temporary lateral folds on each flank, a feature not seen in the other mentioned species [57]. Altogether, these differences suggest that fold formation during dipteran embryogenesis is a complex subject, warranting further investigation to understand its evolutionary mechanisms.

In addition to morphologic and morphogenic differences, variation in the embryonic development timelines can be observed among dipteran species (Fig 8B). For instance, the relative duration of stage 5 in the medfly is more than twice as long as in the fruit fly, scuttle fly and moth midge. This extended duration may stem from a unique aspect of the cellularization process in the medfly. For the fruit fly, a leading model for furrow invagination postulates that a considerable part of the driving force is generated via microtubule motors, which pull the furrow down by attaching to the F-actin cortex via *pav* (PavKLP kinesin) [74]. In contrast, studies in the medfly indicate an absence of F-actin along the cleavage furrow [36], suggesting that an alternative mechanism, potentially one that does not prioritize speed, drives this process in

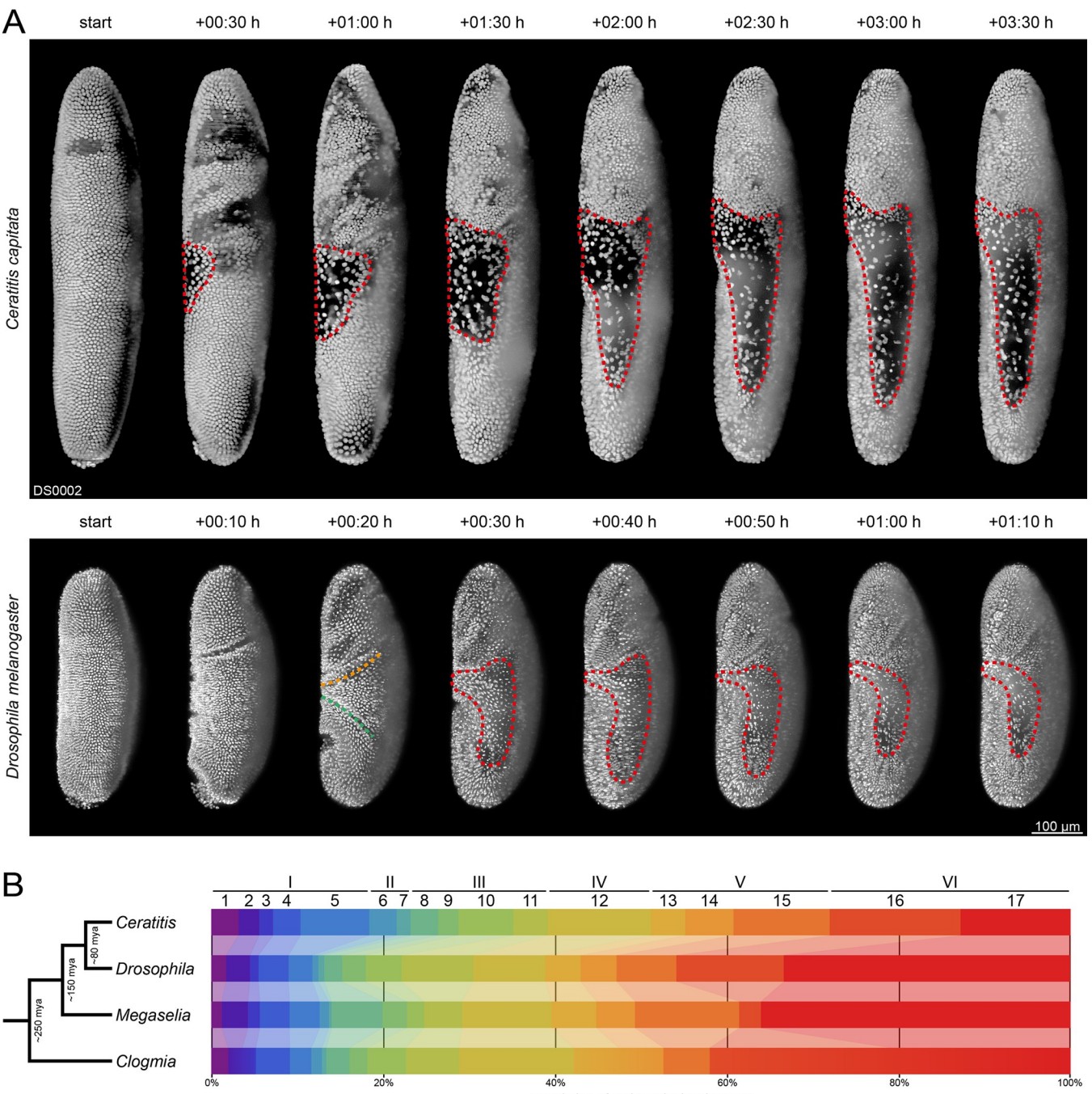

**Fig 8. Comparison of specific aspects in dipteran embryogenesis. (A)** Juxtaposition of extra-embryonic membrane formation in the medfly *Ceratitis capitata* and the fruit fly *Drosophila melanogaster*. In the medfly, the amnioserosa differentiates dorsally as a membrane of widely spaced cells that rearranges as the germband tip migrates anteriorly. In contrast, the primordial amnioserosa cells of the fruit fly remain initially clustered, and the subsequent lateral expansion process occurs without dorsal compression (red dashed lines). The anterior transversal fold (orange dashed line) as well as the posterior (green dashed line) are only found in the fruit fly. The fruit fly data derives from [69]. **(B)** Juxtaposition of relative embryonic development stage duration in four dipteran model organisms. Notably, stage 5, in which cellularization occurs, lasts more than twice as long in *Ceratitis* relative to the other listed species. The *Drosophila melanogaster* data derive from [38], the *Megaselia abdita* data derive from [12], and the *Clogmia albipunctata* data derive from [11]. Please note that in the *Clogmia* staging system, stages 2 and 3, 13 and 14 as well as 16 and 17 are merged (indicated in this figure by the color gradients).

this species. This hypothesis is reinforced by studies in the fruit fly demonstrating that the inhibition of PavKLP impairs force generation, resulting in delayed furrow growth [75].

The previous three paragraphs illustrate how systematically acquired image data can provide insights into the evolutionary history of dipteran development. However an in-depth comparison of differences between the medfly and other insect species is beyond the scope of this work and will be addressed in a separate study.

## Perspective

Our approach–long-term fluorescence live imaging using LSFM–serves as a template for further approaches aimed at characterizing the embryonic development of insect model organisms and establishing comprehensive staging systems that serve as integrative framework for future studies. Although the initial effort is relatively high due to the requirement of one or more suitable transgenic lines, high-quality data can be acquired with only moderate effort when those lines become available. As long-term fluorescence live imaging data from multiple insect species become accessible, morphological structures and the associated morphogenic processes can be thoroughly analyzed within an evolutionary context. Currently, LSFM data for four insect species–the fruit fly [76–79], the scuttle fly [70], the red flour beetle [80–82] and now the medfly [42]–are already available, facilitating comparative research on the embryonic morphogenesis of insects.

## Supporting information

**S1 Table. Methodological and structural summary of selected publications that describe the embryogenesis of twenty-four insect species from ten orders.** The entries are ordered primarily by phylogenetic distance to *Ceratitis capitata* and secondarily alphabetically. (DOCX)

**S2 Table. Detailed description of medfly embryogenesis in chronological order.** Primary stage identifiers are shown in bold. Only time points with clearly identifiable morphogenic changes compared to previous stages are listed. Except for cellularization, we only describe structures and processes for which our data provides morphological and/or dynamic evidence. In the 'developmental process' column, indentions indicate associated processes while parentheses refer to canonical processes summarized in S3 Table. In the 'figures' column, parentheses refer to figures that show the respective process secondarily while curly brackets refer to arrows in the respective figures. (DOCX)

**S3 Table. Overview of ten canonical processes in chronological order.** In the 'figures' column, parentheses refer to figures that show the respective process secondarily. (DOCX)

**S4 Table. Overview of embryonic structures in alphabetical order.** This glossary table provides a definition of all structural terms highlighted in italics throughout this publication. In the 'figures' column, parentheses refer to figures that show the respective process secondarily while curly brackets refer to arrows in the respective figures. (DOCX)

**S5 Table. Selection of structures and canonical processes that are found the fruit fly but have not been described in the medfly.** Our study characterizes the embryogenesis of the medfly mainly on the tissue and cell levels. Thus, we do not describe any subcellular structures

and processes with the exception of nuclei.
(DOCX)

**S1 Fig. Stage-based quick lookup chart of medfly embryogenesis.** The first column indicates embryonic events (Roman numerals), stages (Arabic numerals) and reference to the corresponding figures. The second column shows exemplary Z maximum projections of embryos in a suitable orientation at the given stage. The third column summarizes start, duration, and end of the given stage in hours (blue) and in percent of total development (purple). The fourth column outlines up to four developmental actions during the respective stage. ZA, Z maximum projection with image adjustment.
(TIF)

**S1 Video. Medfly embryogenesis (dataset DS0001) along four perspectives (ventrolateral, dorsolateral, dorsolateral, and ventrolateral) from 02:00 h to 62:00 h with an interval of 00:30 h between the time points.** The video starts at the beginning of stage 2, when the nuclei move to the surface of the egg and stops at the end of stage 17, when the larvae hatch from the egg, showing six embryogenetic events: (I) blastoderm formation, (II) early gastrulation, (III) germband elongation, (IV) germband retraction, (V) dorsal closure and (VI) muscular movement. The video runs at a rate of three frames per second, *i.e.*, 216× faster than the recording time. ZA, Z maximum projection with image adjustment; ven-lat, ventrolateral; dor-lat, dorsolateral.
(AVI)

**S2 Video. Medfly embryogenesis (dataset DS0002) along four perspectives (ventral, lateral, dorsal, and lateral) from 02:00 h to 64:30 h with an interval of 00:30 h between the time points.** The video starts at the beginning of stage 2, when the *zygotic nuclei* migrate to the surface of the egg and stops at the end of stage 17, when the *embryo* completes embryonic development, hatches, and turns into the larva. The video runs at a rate of three frames per second, *i.e.*, 216× faster than the recording time. ZA, Z maximum projection with image adjustment.
(AVI)

**S3 Video. Arivis-based volume rendering of the *head* (dataset DS0009) along the ventrolateral perspective over 25:00 h with an interval of 00:30 h between the time points.** The video starts during stage 13, shortly before the *clypeolabrum* turns from an anterior-dorsal to an anterior-ventral orientation and stops during stage 16, shortly after the *head involution* process is completed. The video runs at a rate of three frames per second, *i.e.*, 216× faster than the recording time.
(AVI)

## Acknowledgments

We thank Maximilian Sandmann and Sven Plath for technical support.

## Author Contributions

**Conceptualization:** Frederic Strobl, Marc F. Schetelig, Ernst H. K. Stelzer.

**Data curation:** Frederic Strobl, Alexander Schmitz.

**Formal analysis:** Frederic Strobl, Alexander Schmitz.

**Funding acquisition:** Frederic Strobl, Ernst H. K. Stelzer.

**Investigation:** Frederic Strobl.

**Methodology:** Frederic Strobl.

**Project administration:** Ernst H. K. Stelzer.

**Resources:** Marc F. Schetelig.

**Software:** Alexander Schmitz.

**Supervision:** Ernst H. K. Stelzer.

**Validation:** Frederic Strobl, Marc F. Schetelig, Ernst H. K. Stelzer.

**Visualization:** Frederic Strobl, Alexander Schmitz.

**Writing – original draft:** Frederic Strobl, Ernst H. K. Stelzer.

**Writing – review & editing:** Frederic Strobl, Marc F. Schetelig, Ernst H. K. Stelzer.

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
