## [Decision Letter · Decision Letter 0]

30 Jul 2024

PONE-D-24-09472A two-level staging system for the embryonic morphogenesis of the Mediterranean fruit fly (medfly) Ceratitis capitataPLOS ONE

Dear Dr. Strobl,

Thank you for submitting your manuscript to PLOS ONE. After careful consideration, we feel that it has merit but does not fully meet PLOS ONE’s publication criteria as it currently stands. Therefore, we invite you to submit a revised version of the manuscript that addresses the points raised during the review process.

We look forward to receiving your revised manuscript.

Kind regards,

Michael Schubert

Academic Editor

PLOS ONE

4. Please update your submission to use the PLOS LaTeX template. The template and more information on our requirements for LaTeX submissions can be found at http://journals.plos.org/plosone/s/latex.

5. We noted in your submission details that a portion of your manuscript may have been presented or published elsewhere. [Imaging data derived from one of our previous studies (Strobl 2022, Scientific Data, https://doi.org/10.1038/s41597-022-01443-x). The Scientific Data journal is specialized on data-only publications without comprehensive data processing, analysis, characterization and interpretation (which is done in this study).] Please clarify whether this [conference proceeding or publication] was peer-reviewed and formally published. If this work was previously peer-reviewed and published, in the cover letter please provide the reason that this work does not constitute dual publication and should be included in the current manuscript.

6. Please provide a complete Data Availability Statement in the submission form, ensuring you include all necessary access information or a reason for why you are unable to make your data freely accessible. If your research concerns only data provided within your submission, please write "All data are in the manuscript and/or supporting information files" as your Data Availability Statement.

7. Please amend either the title on the online submission form (via Edit Submission) or the title in the manuscript so that they are identical.

8. We notice that your supplementary figures and tables are included in the manuscript file. Please remove them and upload them with the file type 'Supporting Information'. Please ensure that each Supporting Information file has a legend listed in the manuscript after the references list.

Reviewers' comments:

Reviewer's Responses to Questions

**Comments to the Author**

1. Is the manuscript technically sound, and do the data support the conclusions?

Reviewer #1: Yes

Reviewer #2: Yes

2. Has the statistical analysis been performed appropriately and rigorously? 

Reviewer #1: Yes

Reviewer #2: Yes

3. Have the authors made all data underlying the findings in their manuscript fully available?

Reviewer #1: Yes

Reviewer #2: No

4. Is the manuscript presented in an intelligible fashion and written in standard English?

Reviewer #1: Yes

Reviewer #2: Yes

5. Review Comments to the Author

Reviewer #1: This paper by Strobl et al. reports on the morphogenetic processes in the embryo of the Mediterranean fruit fly, Ceratitis capitata, utilizing live imaging of transgenic animals labeled with nuclear fluorescence. The authors employed cutting-edge in toto imaging technologies, including dual-axis light sheet fluorescence microscopy, to observe whole embryos throughout nearly the entire stage of embryogenesis. Preliminary findings on image acquisition were previously published (https://www.nature.com/articles/s41597-022-01443-x).

The article presents an extensive quantitative analysis of Ceratitis embryogenesis, based on the independent recording of nine embryos. Image segmentation techniques were utilized to obtain digitized images of each nucleus and the outline of the embryo's outer surface. From these images, the authors compiled a comprehensive set of quantitative data on nuclear size, density, and morphogenetic movements. This data is meticulously summarized in numerous figures and tables, proving to be an invaluable resource for researchers studying Ceratitis. Additionally, the dataset provides comparative insights beneficial to a broader group of scientists working on Drosophila and other insects.

Historically, the study of developmental processes in embryogenesis has relied on detailed descriptive references such as those provided by pioneers like Don Poulson (1937, 1950), Campos-Ortega and Hartenstein (1985), and Wieschaus and Nusslein-Volhard (1986) for Drosophila. This work by Strobl singularly achieves this for Ceratitis, with an exceptional precision in spatiotemporal details. The application of live-imaging technology in this study sets a new standard in morphological research.

I would like to express my full appreciation for the extensive effort and scientific rigor demonstrated by the authors.

Typo: line 279 laterad -> lateral

Reviewer #2: The manuscript by Strobl et al. describes the embryonic development of Ceratitis capitata, a widely studied species of great agricultural interest and informative phylogenetic position in the dipteran phylogeny. Based on live imaging datasets of several individuals acquired with lightsheet microscopy, the authors generate a comprehensive staging system detailing the embryonic events from cleavage to hatching. This includes quantitative analyses of temporal dynamics and variability between individuals, which is extremely valuable (and rare) in developmental studies. The methods are thoroughly described and, despite the complexity of the data, the reporting of the results is clear and comprehensible. Textual descriptions are well written, figures and data visualizations are intuitive, and the different developmental tables organized by chronology and structures are also incredibly helpful. Overall, this is one of the most systematic and comprehensive developmental staging systems that I have read recently and exemplifies how much valuable information can be derived from in toto live imaging datasets. The work will serve as an exceptional resource for the dipteran community and for future comparative studies on dipteran and insect evo-devo.

I have only one main suggestion regarding the discussion. The authors mention that an in-depth comparison between the medfly and other insects will be the subject of a subsequent study. However, I believe that including a general comparison between Ceratitis and Drosophila would greatly benefit the current manuscript. While this is accomplished to some extent in the discussion of extra-embryonic tissues and in Supplementary Tables 4 and 5, it could be more effectively summarized in a new discussion figure highlighting the key differences between the two species (e.g., timing of embryonic events, morphogenetic differences, presence/absence of traits, etc.). Having this concise overview with the most notable differences would be a proof-of-concept supporting the main motivation behind the work—providing the grounds for comparative studies on developmental evolution. In principle, this would not take away from a future, more in-depth comparative study across insects, but I leave to the authors the decision to follow on this suggestion or not.

Please find below some additional minor points and comments.

Introduction

* Pg3 Ln71: There is also a short paper from the "Proceedings of the 6th International Fruit Fly Symposium" on the "Early developmental stages of Ceratitis capitata embryos" that provides a basic description of the medfly embryonic stages until germ band elongation and compares the timing of developmental events to Drosophila. I believe it should also be included in this sentence mentioning previous works. The citation is: Stefani RN, Selivon D & Perondini ALP (2004) Early developmental stages of Ceratitis capitata embryos. In Proceedings of the 6th International Symposium on fruit flies of economic importance, Stellenbosch, South Africa, 6-10 May 2002 (PDF available from the IEAE repository: https://nucleus.iaea.org/sites/naipc/twd/Documents/6thISFFEI_Proceedings/STEFANI.pdf)

Methods

* Pg5 Ln98: I applaud the authors for publishing and depositing the raw lightsheet datasets beforehand.

* Pg5 Ln102-105: The wording "from which dataset X derives" makes theses sentences difficult to read. I suggest using a more direct form like "The embryo of dataset DS0006 did not develop..."

* Pg5 Ln121 and Pg6 Ln137,149: While the image processing steps are described in great detail, the software programs used for these routines are not mentioned. Please cite them in the methods.

* Pg6 Ln149: In addition, please make any source code and measurement data used in the manuscript available as supporting information or in a data repository.

Results

* Pg7 Ln178: It might be helpful to mention the main embryonic processes happening during these deviations. For example, "...at the transition from stage 10 to stage 11 (during germ band elongation)...". Also, do you have a hypothesis about the reasons for these deviations? Are they technical or biological? This is an interesting observation that should be included in the discussion as an example of how quantitative data can be informative for understanding developmental processes.

* Pg9 Ln230: The text states that the "yolk begins withdrawing" and that made me confused as I thought that the yolk inside the embryo was moving inwards. But based on Fig2C, I believe you are referring to the membrane of the embryo that is withdrawing relative to the vitelline. Please edit the text to clarify.

* Pg19 Ln528 (Fig2C): I assume the red dashed line is the vitelline envelope, but it is not annotated in the panel or in the figure legend. Actually, in most panels where the red dashed line is used to label the vitelline, there's no description (see 3B and 6B), only when it is marking other structures. Please include the description in the legends of the respective panels.

* Pg11 Ln305: Replace "}" with ")"

* Pg12 Ln353: I was slightly confused with this sentence going "back in time" to the beginning of stage 12 when the previous paragraph had finished at the end of stage 12. Re-rewording or a short topic sentence should be sufficient (e.g., "During germband retraction, we observe different dynamics of abdomen and head withdrawal."). In addition, Fig5A does not have any annotations pointing to the abdomen withdrawal. It doesn't need a full crop, as for the head, but a small arrow pointing to the abdomen withdrawal at the posterior tip would make it easier for the reader to find it.

Discussion

* Pg15 Ln436: To avoid confusion, I suggest reminding the reader that "fruit fly" here refers to Drosophila. This is clear in the introduction, but it is worth being explicit at the beginning of the discussion as well.

* Pg15 Ln436-446: The comparison between the medfly and fruit fly amnioserosa is important, but the text needs to be edited to improve clarity. Both morphogenetic processes are described in length, but the key difference between the two was not clear to me until I read Ln451 from the subsequent paragraph. If I understood correctly, you want to highlight that in the medfly, the differentiation of the amnioserosa cells happens early, before germ band extension, while in Drosophila, the germ band is already extended when the lateral expansion occurs. I propose re-writing this part focusing on GB elongation (not retraction) to something like: "Both the medfly and fruit fly form the amnioserosa dorsally, but the relative timing between amnioserosa differentiation and germ band extension is different between the two species. While in the medfly (...), in the fruit fly (...)."

* Pg15 Ln446: I find transient epithelial folds fascinating. However, the mention of dorsal folds and lateral folds is somewhat disconnected from the main subject regarding the temporal differences in amnioserosa/germband differentiation. Are you suggesting an association between premature amnioserosa and a lack of dorsal folds (or vice versa)? Or that lateral folds might be a "compensating" mechanism in the Queensland fly? In any case, I recommend stating more explicitly why these observations are interesting and how they connect to the extra-embryonic comparative analysis; after all, there are not many hypotheses about the evolution of dorsal folds.

* Pg15 Ln459: In fact, I feel the discussion could even be slightly expanded to highlight a few more key differences between Ceratitis and Drosophila embryogenesis developmental trajectories. While the authors mention that a more detailed comparative analysis will be the subject of another manuscript, having one discussion figure with schematic overview of the differences in traits mentioned in the discussion would give strong support to one of the key motivations of the work—that such detailed embryogenetic data is greatly informative for understanding developmental evolution. But I leave this decision to the authors.

* Pg15 Ln460: The technical part of the discussion raises good points and is well-balanced. However, I feel that it would fit best as the starting point for the discussion rather than the end. I suggest beginning the discussion with what your approach brings to the field, and then discussing the biological implications of your findings. Moving the biological discussion to the end feels more natural to me and would highlight the value of careful descriptive and comparative approaches to understanding the evolutionary history of dipteran development.

Supplementary

* Pg35 Ln617, Pg41 Ln630 (ventral mesectoderm), Pg44 Ln640 (anterior transversal furrow, mesoderm development, posterior transversal furrow): Error! Reference source not found.

* Pg42 Ln634 (peripheral migration, cellularization), Pg43 Ln635 (digestive system formation): There is a line break after "(Supplementary Table 1"

* Pg33 Ln607: Include Ceratitis in this table for completion, citing the current work.

Best regards,

Bruno C. Vellutini

6. PLOS authors have the option to publish the peer review history of their article (what does this mean?). If published, this will include your full peer review and any attached files.

Reviewer #1: No

Reviewer #2: **Yes: **Bruno Cossermelli Vellutini

---

## [Author Response · Author response to Decision Letter 0]

13 Nov 2024

Editorial Comments

Concern E-1

Please ensure that your manuscript meets PLOS ONE's style requirements, including those for file naming. (…)

Answer E-1

We ensured that our manuscript meets the style requirements.

Concern E-2

We suggest you thoroughly copyedit your manuscript for language usage, spelling, and grammar. (…)

Answer E-2

The manuscript was professionally copyedited. Please note that there are several changes (tracked with the respective function) throughout the manuscript.

Concern E-3

We note that the grant information you provided in the ‘Funding Information’ and ‘Financial Disclosure’ sections do not match.

Answer E-3

The information in both sections match now.

Concern E-4

Please update your submission to use the PLOS LaTeX template.

Answer E-4

We do not use LaTeX and would like to resubmit in the DOCX format.

Concern E-5

We noted in your submission details that a portion of your manuscript may have been presented or published elsewhere. [Imaging data derived from one of our previous studies (Strobl 2022, Scientific Data, https://doi.org/10.1038/s41597-022-01443-x). The Scientific Data journal is specialized on data-only publications without comprehensive data processing, analysis, characterization and interpretation (which is done in this study).] Please clarify whether this [conference proceeding or publication] was peer-reviewed and formally published. If this work was previously peer-reviewed and published, in the cover letter please provide the reason that this work does not constitute dual publication and should be included in the current manuscript.

Answer E-5

We clarified in the manuscript that the cited publication is peer-reviewed and added an explanation to the cover letter as requested.

Concern E-6

Please provide a complete Data Availability Statement in the submission form, ensuring you include all necessary access information or a reason for why you are unable to make your data freely accessible. If your research concerns only data provided within your submission, please write "All data are in the manuscript and/or supporting information files" as your Data Availability Statement.

Answer E-6

We do now provide a complete Data Availability Statement in the submission form.

Concern E-7

Please amend either the title on the online submission form (via Edit Submission) or the title in the manuscript so that they are identical.

Answer E-7

The title in the online submission from and the manuscript match now.

Concern E-8

We notice that your supplementary figures and tables are included in the manuscript file. Please remove them and upload them with the file type 'Supporting Information'. Please ensure that each Supporting Information file has a legend listed in the manuscript after the references list.

Answer E-8

Done.

Concern E-9

Answer E-9

We exhaustively revised the reference list to ensure that it is complete and correct, and to align it with the PLoS One formatting guidelines.

Comments to the Author (only comments with implications for the revision)

Concern C-3

Have the authors made all data underlying the findings in their manuscript fully available?

Reviewer #1: Yes

Reviewer #2: No

Answer C3

We assume that the “No” from Reviewer #2 here refers to their statement on source code availability that we address in Concern #2-5 and Answer #2-5.

Reviewer #1

Concern #1-1

Typo: line 279 laterad -> lateral).

Answer #1-1

We corrected this typo.

Reviewer #2

Concern #2-1

I have only one main suggestion regarding the discussion. The authors mention that an in-depth comparison between the medfly and other insects will be the subject of a subsequent study. However, I believe that including a general comparison between Ceratitis and Drosophila would greatly benefit the current manuscript. While this is accomplished to some extent in the discussion of extra-embryonic tissues and in Supplementary Tables 4 and 5, it could be more effectively summarized in a new discussion figure highlighting the key differences between the two species (e.g., timing of embryonic events, morphogenetic differences, presence/absence of traits, etc.). Having this concise overview with the most notable differences would be a proof-of-concept supporting the main motivation behind the work—providing the grounds for comparative studies on developmental evolution. In principle, this would not take away from a future, more in-depth comparative study across insects, but I leave to the authors the decision to follow on this suggestion or not.

Answer #2-1

Reviewer #2 raises an important concern here, with which we agree. Following their suggestion, we designed an additional figure (Figure 8) that (i) illustrates morphogenetic differences exemplarily via a side-by-side comparison of Drosophila- and Ceratitis-derived microscopy data (Figure 8A) and (ii) compares the relative timing of the 17 embryonic stages between Drosophila, Ceratitis, Megaselia and Clogmia (Figure 8B). The “Extra-embryonic membranes in the medfly and other insect species” section was slightly adjusted to incorporate the new figure.

Concern #2-2

* Pg3 Ln71: There is also a short paper from the "Proceedings of the 6th International Fruit Fly Symposium" on the "Early developmental stages of Ceratitis capitata embryos" that provides a basic description of the medfly embryonic stages until germ band elongation and compares the timing of developmental events to Drosophila. I believe it should also be included in this sentence mentioning previous works. The citation is: Stefani RN, Selivon D & Perondini ALP (2004) Early developmental stages of Ceratitis capitata embryos. In Proceedings of the 6th International Symposium on fruit flies of economic importance, Stellenbosch, South Africa, 6-10 May 2002 (PDF available from the IEAE repository: https://nucleus.iaea.org/sites/naipc/twd/Documents/6thISFFEI_Proceedings/STEFANI.pdf)

Answer #2-2

We thank Reviewer #2 for pointing out this reference, which we probably overlooked due to the reason that it is not indexed on NCBI/PubMed. We are not sure whether this study has been peer-reviewed; we however assume that is has not since it has been published as part of the Proceedings of the 6th International Fly Symposium. We thoroughly read the publication and concluded that the quality is sufficient to be cited. Hence, we do now cite this paper in the introduction.

Since the publication is not peer-reviewed, we will leave the final decision to the PLoS One editorial team whether the new citation may stay or should be removed again.

Concern #2-3

* Pg5 Ln98: I applaud the authors for publishing and depositing the raw lightsheet datasets beforehand.

Answer #2-3

We appreciate the positive remark from Reviewer #2 and would like to use the opportunity to point again at Concern E-5, Answer E-5 and the respective explanation in the revision cover letter – we believe that modern image-heavy studies may yield so much valuable data that in certain cases such as this one, a split into a “data publication” and a “biological insights publication” is more than reasonable. Packing everything into just one publication would feel cluttered. 

Concern #2-3

* Pg5 Ln102-105: The wording "from which dataset X derives" makes theses sentences difficult to read. I suggest using a more direct form like "The embryo of dataset DS0006 did not develop..."

Answer #2-3

We changed the sentence structures according to the suggestion.

Concern #2-4

* Pg5 Ln121 and Pg6 Ln137,149: While the image processing steps are described in great detail, the software programs used for these routines are not mentioned. Please cite them in the methods.

Answer #2-4

We added an additional short paragraph called “Fluorescence microscopy data processing and analysis” to the Materials and methods sections that references the software that we used while also linking to a Zenodo-stored data processing and analysis package that contains processed image data, the Mathematica notebooks, as well as intermediate measurement and visualization files.

Concern #2-5

* Pg6 Ln149: In addition, please make any source code and measurement data used in the manuscript available as supporting information or in a data repository.

Answer #2-5

We assume that this concern is associated with the “No”-Remark regarding the question whether all data has been made available (Concern C-3). Please see Concern #2-4 and Answer #2-4.

Concern #2-6

* Pg7 Ln178: It might be helpful to mention the main embryonic processes happening during these deviations. For example, "...at the transition from stage 10 to stage 11 (during germ band elongation)...". Also, do you have a hypothesis about the reasons for these deviations? Are they technical or biological? This is an interesting observation that should be included in the discussion as an example of how quantitative data can be informative for understanding developmental processes.

Answer #2-6

We added short remainders in brackets at the indicated positions as suggested. We share Reviewer #2’s opinion that this is a very interesting observation and further added several sentences illustrating our interpretation regarding these deviations to the “Quantitative analysis” section of the discussion.

Concern #2-7

* Pg9 Ln230: The text states that the "yolk begins withdrawing" and that made me confused as I thought that the yolk inside the embryo was moving inwards. But based on Fig2C, I believe you are referring to the membrane of the embryo that is withdrawing relative to the vitelline. Please edit the text to clarify.

Answer #2-7

We clarify the mentioned passage.

Concern #2-8

* Pg19 Ln528 (Fig2C): I assume the red dashed line is the vitelline envelope, but it is not annotated in the panel or in the figure legend. Actually, in most panels where the red dashed line is used to label the vitelline, there's no description (see 3B and 6B), only when it is marking other structures. Please include the description in the legends of the respective panels.

Answer #2-8

As suggested, we added an explanation to all of the mentioned figure captions (and also in occasions within the running text).

Concern #2-9

* Pg11 Ln305: Replace "}" with ")".

Answer #2-9

Done.

Concern #2-10

* Pg12 Ln353: I was slightly confused with this sentence going "back in time" to the beginning of stage 12 when the previous paragraph had finished at the end of stage 12. Re-rewording or a short topic sentence should be sufficient (e.g., "During germband retraction, we observe different dynamics of abdomen and head withdrawal."). In addition, Fig5A does not have any annotations pointing to the abdomen withdrawal. It doesn't need a full crop, as for the head, but a small arrow pointing to the abdomen withdrawal at the posterior tip would make it easier for the reader to find it.

Answer #2-10

Our intention for the running text is to follow processes over long time periods as stated on page 8 (“The running text is divided into six sections that relate to the six embryogenetic events. Each section is accompanied by a comprehensively illustrated figure. Complex processes that involve multiple structures are described over longer periods. Only medfly-related studies are referenced within the running text. Numbers in curly brackets reference to arrows in the respective figures.”). Hence, here, the first paragraph concerns the germband dynamics, the second jumps back in time a little and describes the withdrawals. This stands in contrast to Supplementary Table 2, which is strictly chronological. However, we think that reviewer #2 is correct that a brief introduction sentence is missing, which we added now.

We also added additional arrows regarding abdomen withdrawal to Fig. 5A that are cited within the running text and the figure description.

Concern #2-11

* Pg15 Ln436: To avoid confusion, I suggest reminding the reader that "fruit fly" here refers to Drosophila. This is clear in the introduction, but it is worth being explicit at the beginning of the discussion as well.

Answer #2-11

For consistency and to improve readability, we do now mention the scientific name directly after common name for each species that we mention in the discussion at the first occurrence.

Concern #2-12

* Pg15 Ln436-446: The comparison between the medfly and fruit fly amnioserosa is important, but the text needs to be edited to improve clarity. Both morphogenetic processes are described in length, but the key difference between the two was not clear to me until I read Ln451 from the subsequent paragraph. If I understood correctly, you want to highlight that in the medfly, the differentiation of the amnioserosa cells happens early, before germ band extension, while in Drosophila, the germ band is already extended when the lateral expansion occurs. I propose re-writing this part focusing on GB elongation (not retraction) to something like: "Both the medfly and fruit fly form the amnioserosa dorsally, but the relative timing between amnioserosa differentiation and germ band extension is different between the two species. While in the medfly (...), in the fruit fly (...)."

Answer #2-12

Inspired by the suggestions, we edited the respective paragraph in the discussion. The message that we want to provide to the dedicated reader should be clearer now.

Concern #2-13

* Pg15 Ln446: I find transient epithelial folds fascinating. However, the mention of dorsal folds and lateral folds is somewhat disconnected from the main subject regarding the temporal differences in amnioserosa/germband differentiation. Are you suggesting an association between premature amnioserosa and a lack of dorsal folds (or vice versa)? Or that lateral folds might be a "compensating" mechanism in the Queensland fly? In any case, I recommend stating more explicitly why these observations are interesting and how they connect to the extra-embryonic comparative analysis; after all, there are not many hypotheses about the evolution of dorsal folds.

Answer #2-13

We would like to mention that we share Reviewer #2’s enthusiasm for epithelial folds. With regard to their comment, we separated the paragraph into paragraphs, the first one elaborating on extra-embryonic membranes, the second one discussing the formation of epithelial folds. We also added a paragraph regarding the timing of embryonic development to this section of the discussion (cf. Concern #2-14). The section also references the new figure, Figure 8. Since we discuss three distinct subjects now, we changed the name of the section from “Extra-embryonic membranes in the medfly and other insect species” to “Developmental differences between the medfly and other insect species”.

Concern #2-14

* Pg15 Ln459: In fact, I feel the discussion could even be slightly expanded to highlight a few more key differences between Ceratitis and Drosophila embryogenesis de

---

## [Decision Letter · Decision Letter 1]

25 Nov 2024

PONE-D-24-09472R1A two-level staging system for the embryonic morphogenesis of the Mediterranean fruit fly (medfly) Ceratitis capitataPLOS ONE

Dear Dr. Strobl,

Thank you for submitting your manuscript to PLOS ONE. After careful consideration, we feel that it has merit but does not fully meet PLOS ONE’s publication criteria as it currently stands. Therefore, we invite you to submit a revised version of the manuscript that addresses the minor points raised during the review process.

We look forward to receiving your revised manuscript.

Kind regards,

Michael Schubert

Academic Editor

PLOS ONE

Journal Requirements:

Reviewers' comments:

Reviewer's Responses to Questions

**Comments to the Author**

1. If the authors have adequately addressed your comments raised in a previous round of review and you feel that this manuscript is now acceptable for publication, you may indicate that here to bypass the “Comments to the Author” section, enter your conflict of interest statement in the “Confidential to Editor” section, and submit your "Accept" recommendation.

Reviewer #1: All comments have been addressed

Reviewer #2: All comments have been addressed

2. Is the manuscript technically sound, and do the data support the conclusions?

Reviewer #1: Yes

Reviewer #2: Yes

3. Has the statistical analysis been performed appropriately and rigorously? 

Reviewer #1: Yes

Reviewer #2: Yes

4. Have the authors made all data underlying the findings in their manuscript fully available?

Reviewer #1: Yes

Reviewer #2: Yes

5. Is the manuscript presented in an intelligible fashion and written in standard English?

Reviewer #1: Yes

Reviewer #2: Yes

6. Review Comments to the Author

Reviewer #1: AS for the first submission of this manuscript, this work represents a tour-de-force effotrt to document the standard time table and morphology of Ceratitis embryogenesis. This will be one of the classic reading for entomologists.

Reviewer #2: The authors have fully addressed my comments, and the revised manuscript is finely polished. I only found a couple of wording issues in the text:

Pg22 Ln548: Extra word "the available image data have are limited"

Pg24 Ln592: Difficult to read "dipteran species such as the scuttle fly [12,14–16] and the moth midge [11] appear to morphogenically specify their extra-embryonic membranes also not before germband elongation." Suggestion: "also appear to form their extra-embryonic membranes after germband elongation"

Best regards,

Bruno C. Vellutini

7. PLOS authors have the option to publish the peer review history of their article (what does this mean?). If published, this will include your full peer review and any attached files.

Reviewer #1: No

Reviewer #2: **Yes: **Bruno Cossermelli Vellutini

---

## [Author Response · Author response to Decision Letter 1]

27 Nov 2024

Concern #2-Round2-1

* Pg22 Ln548: Extra word "the available image data have are limited"

Answer #2-Round2-1

We corrected the text passage.

Concern #2-Round2-2

* Pg24 Ln592: Difficult to read "dipteran species such as the scuttle fly [12,14–16] and the moth midge [11] appear to morphogenically specify their extra-embryonic membranes also not before germband elongation." Suggestion: "also appear to form their extra-embryonic membranes after germband elongation"

Answer #2-Round2-2

We corrected the text passage as mentioned by Reviewer #2, with the small change that we changed “after germband elongation” to “after early gastrulation” in their suggestion, as this is the correct simplification from the original “not before germband elongation”.

---

## [Editor Report · Decision Letter 2]

11 Dec 2024

A two-level staging system for the embryonic morphogenesis of the Mediterranean fruit fly (medfly) Ceratitis capitata

PONE-D-24-09472R2

Dear Dr. Strobl,

We’re pleased to inform you that your manuscript has been judged scientifically suitable for publication and will be formally accepted for publication once it meets all outstanding technical requirements.

Kind regards,

Michael Schubert

Academic Editor

PLOS ONE

---

## [Editor Report · Acceptance letter]

16 Dec 2024

PONE-D-24-09472R2 

PLOS ONE

Dear Dr. Strobl, 

I'm pleased to inform you that your manuscript has been deemed suitable for publication in PLOS ONE. Congratulations! Your manuscript is now being handed over to our production team.

Kind regards, 

on behalf of

Dr. Michael Schubert 

Academic Editor

PLOS ONE